# FrameOracle: Learning What to See and How Much to See in Videos

**Chaoyu Li** [1 2 †]  **Tianzhi Li** [1 3 †]  **Fei Tao** [1 ‡]  **Zhenyu Zhao** [1]  **Ziqian Wu** [1]  **Maozheng Zhao** [1]  **Juntong Song** [1]
**Cheng Niu** [1]  **Pooyan Fazli** [2]

## Abstract

Vision-language models (VLMs) advance video understanding but operate under tight computational budgets, making performance dependent on selecting a small, high-quality subset of frames. Existing frame sampling strategies, such as uniform or fixed-budget selection, fail to adapt to variations in content density or task complexity. To address this, we present **FrameOracle**, a lightweight, plug-and-play module that predicts both (1) which frames are most relevant to a given query and (2) how many frames are needed. FrameOracle is trained via a curriculum that progresses from weak proxy signals, such as cross-modal similarity, to stronger supervision with **FrameOracle-41K**, the first large-scale VideoQA dataset with validated keyframe annotations specifying minimal sufficient frames per question. Extensive experiments across five VLMs and six benchmarks show that FrameOracle reduces 16-frame inputs to an average of 10.4 frames without accuracy loss. When starting from 64-frame candidates, it reduces inputs to 13.9 frames on average while improving accuracy by 1.5%, achieving state-of-the-art efficiency–accuracy trade-offs for scalable video understanding. Our project is available here: https://people-robots.github.io/frameoracle.

## 1. Introduction

Rapid advances in large language models (LLMs) (Stiennon et al., 2020; Gao et al., 2022; Yang et al., 2024) have enabled vision-language models (VLMs) to combine visual understanding with strong linguistic reasoning (Zhang et al., 2025e; Bai et al., 2025b; Zhang et al., 2025a). As a result,

VLMs are highly effective for complex video tasks such as question answering (Zhang et al., 2023; Zhao et al., 2024; Xiao et al., 2025; Li et al., 2025b), summarization (Hua et al., 2025; Lee et al., 2025b), and instruction following (Ren et al., 2024; Qian et al., 2024). In practice, however, these models must operate under tight computational budgets, limiting the number of video frames they can process. Under such constraints, performance depends on selecting a small, high-quality subset of frames. Most existing VLMs rely on simple strategies, such as uniform sampling at a fixed frame rate or selecting a fixed number of frames (*keyframes*). While easy to implement, these approaches fail to adapt to variations in content density or task complexity, often missing salient moments in long videos while introducing redundant and distracting frames in shorter ones.

To mitigate this issue, a growing body of work explores keyframe selection methods (Park et al., 2026; Zhang et al., 2025d), which aim to preserve semantic content while reducing redundancy. However, most existing approaches assume a fixed number of keyframes, overlooking the fact that the optimal frame budget varies across videos and queries. A few recent methods introduce adaptive frame selection, but their adaptivity remains limited. Some require joint training with a specific VLM backbone (Buch et al., 2025), restricting transferability, while others use heuristics at inference rather than learning frame selection end-to-end (Yu et al., 2025). These limitations raise the question: *Can frame selection be designed as a plug-and-play learnable module that determines which frames to select ("what to see") and how many ("how much to see") for a given query?*

To answer this question, we introduce **FrameOracle**, a lightweight, backbone-agnostic frame selector that can be integrated with any VLM. Unlike prior methods that fix the number of frames or require co-training with a specific backbone, FrameOracle jointly predicts (1) the importance of each frame relative to the query and (2) how many frames to retain. It is trained progressively, starting with weak proxy signals and refined with stronger supervision from our new dataset, **FrameOracle-41K**, the first large-scale VideoQA dataset with keyframe annotations specifying the minimal frames required to answer each question. To our knowledge, no dataset provides ground-truth keyframes for VideoQA. FrameOracle adapts its selections to both video content and

---

[†]Work done while interning at NewsBreak [‡]Project lead
[1]NewsBreak [2]Arizona State University [3]Carnegie Mellon University. Correspondence to: Chaoyu Li <chaoyuli@asu.edu>, Fei Tao <fei.tao@newsbreak.com>.

*Proceedings of the 43rd International Conference on Machine Learning*, Seoul, South Korea. PMLR 306, 2026. Copyright 2026 by the author(s).

query, functioning seamlessly as a preprocessing module for downstream VLMs.

In summary, our contributions are as follows:

- We introduce **FrameOracle**, a plug-and-play frame selector that predicts which frames to see and how many frames to retain for each video and query.

- We present **FrameOracle-41K**, the first large-scale VideoQA dataset with keyframe annotations specifying the minimal frames required to answer each question.

- We conduct experiments across five VLMs and six benchmarks, showing that FrameOracle reduces 16-frame inputs to 10.4 frames on average without accuracy loss, and reduces 64-frame inputs to 13.9 frames while improving accuracy by 1.5%, achieving state-of-the-art efficiency–accuracy trade-offs.

## 2. Related Work

**Keyframe Selection for Video Understanding.** Most existing keyframe selection methods assume a fixed frame budget: they rank candidate frames by visual–linguistic relevance or temporal salience and then retain the top-$k$ subset, a paradigm adopted by methods such as KeyVideoLLM (Liang et al., 2024), BOLT (Liu et al., 2025), MDP3 (Sun et al., 2025), and ReFoCUS (Lee et al., 2025a). Beyond this fixed-budget paradigm, adaptive frame selection has been explored in both classical video recognition and recent VLM-based settings. Early work, such as AdaFrame (Wu et al., 2019), proposes a learned policy to dynamically decide how many frames to process for video classification. More recently, adaptive selection has been revisited in the context of video-language reasoning. These approaches fall into two categories. The first are agent-based methods, in which large models act as decision-makers, iteratively analyzing videos. For instance, VCA (Yang et al., 2025) combines curiosity-driven exploration with tree search to identify informative segments, while AKeyS (Fan et al., 2025) leverages a language agent to heuristically expand video segments and decide both which frames to retain and when to stop. However, such methods are computationally expensive due to repeated agent calls. The second category comprises approaches that require co-training with a specific VLM backbone (Buch et al., 2025; Yu et al., 2023; Guo et al., 2025), which restricts their portability. In contrast, FrameOracle is adaptive, lightweight, and model-agnostic: it learns to jointly predict which frames are relevant and how many to retain, while remaining plug-and-play across diverse VLMs.

**Datasets and Supervision for Video-Language Models.** Progress in video-language reasoning has been driven by large-scale datasets such as LLaVA-Video-178K (Zhang

et al., 2025e), ShareGPT4Video (Chen et al., 2024b), VideoRefer (Yuan et al., 2025), and VideoPASTA (Kulkarni & Fazli, 2025), which cover diverse scenarios and support both short- and long-form understanding. However, most of these datasets provide supervision only at the answer level, leaving the underlying evidence unannotated. In the absence of frame-level labels, keyframe selection methods typically rely on proxy signals, such as leave-one-out degradation or heuristic scoring. A few benchmarks, such as TVQA+ (Lei et al., 2020), ReXTime (Chen et al., 2024a), and HourVideo (Chandrasegaran et al., 2024), move toward span-level annotations, but none supply labels for both the indices of keyframes and the minimal sufficient number of frames needed to answer a question. FrameOracle-41K is the first dataset to provide explicit keyframe annotations for video–question pairs, offering high-quality supervision for both training and evaluation of adaptive frame selectors.

## 3. FrameOracle-41K Dataset

We introduce FrameOracle-41K, the first VideoQA dataset with keyframe annotations that specify the minimal set of frames required to answer each question. The corpus contains 40,992 video-question pairs covering diverse scenes and durations. In contrast to existing VideoQA datasets, which provide only ground-truth answers and, in some cases, approximate temporal regions of the video containing the answer, FrameOracle-41K records, for each instance, the minimal number of frames needed to answer the question, along with the keyframes that provide the necessary evidence. Below, we describe our data generation pipeline, the verification and filtering procedures used to retain high-quality data, and the key statistics of the dataset.

### 3.1. Data Gathering and Processing

All video-question pairs in FrameOracle-41K are sourced from LLaVA-Video-178K (Zhang et al., 2025e), a large-scale dataset covering a wide range of scenarios and activities. From this corpus, we first select nearly 100K videos, each 2–3 minutes long, balancing sufficient temporal context with manageable annotation effort. The final dataset is created through a two-stage process (Figure 1). *Stage I (agent-based keyframe mining)* automatically extracts candidate keyframes using a multimodal agent that iteratively explores each video and assigns frame-level relevance scores. *Stage II (filtering and verification)* first filters candidate keyframes by discarding frames with low relevance scores and then retains only those video–question pairs for which three independent VLMs can correctly answer the question using the remaining keyframes. This ensures that the remaining keyframes contain the necessary evidence to answer each question. We further conduct a human verification on 4,000 randomly sampled instances, achieving an inter-annotator

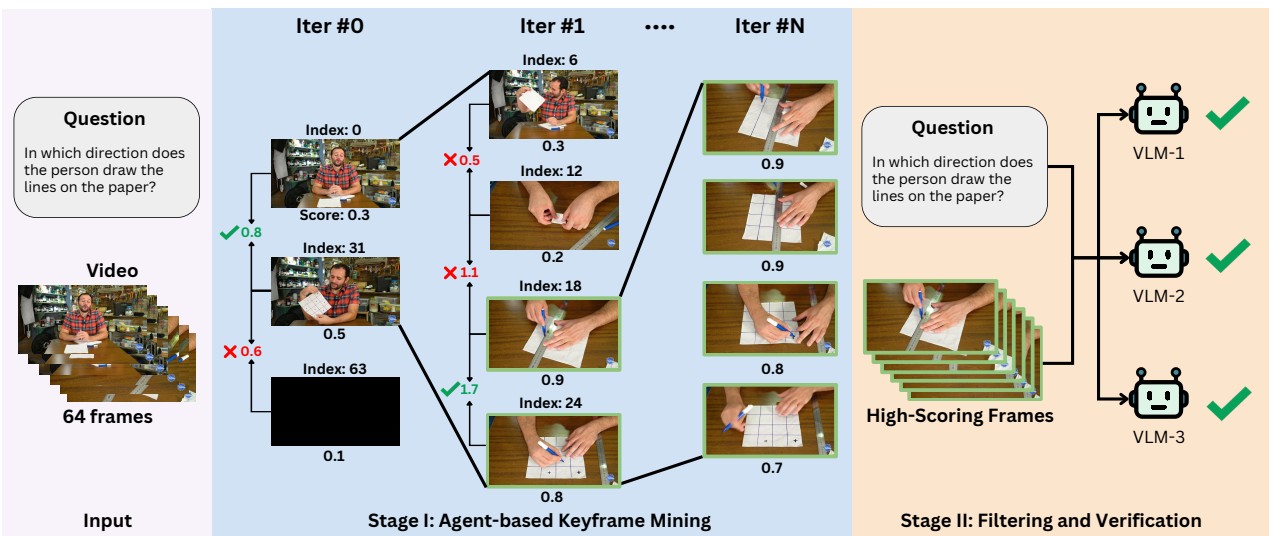

*Figure 1.* **FrameOracle-41K data generation pipeline.** Stage I (agent-based keyframe mining) iteratively explores each video using a multimodal agent, ultimately returning a predicted answer with confidence and relevance scores for all visited frames. Stage II (filtering and verification) first discards frames with low relevance scores and then verifies sufficiency by requiring three independent VLMs to answer correctly using only the remaining keyframes.

agreement of 94% and a verified accuracy of 93.3%. This confirms the reliability of the automatically generated annotations. Prompts for data generation, an example dataset entry, the human verification details, and dataset visualizations are shown in Appendices C, B, E, and H, respectively.

**Stage I: Agent-based Keyframe Mining.** Starting from a uniformly sampled set of 64 frames, we employ an agent built on Qwen2.5-VL-72B API (Bai et al., 2025b) to iteratively explore the video conditioned on the given question. In the first iteration, the agent evaluates three anchor frames (indices 0, 31, and 63), assigns each a relevance score, and attempts to answer the question with an associated confidence estimate. It then compares the summed relevance scores of the two adjacent anchor pairs (0 + 31 vs. 31 + 63) and selects the higher-scoring temporal segment for further exploration. In subsequent iterations, the agent uniformly samples four new anchor frames within the selected segment, repeats relevance scoring and answer prediction, and again selects the sub-segment with the highest aggregated relevance. This coarse-to-fine procedure iteratively narrows the temporal window of interest. The process terminates when the agent reaches a confident answer or when all 64 candidate frames have been evaluated. At the end of Stage I, the agent outputs the set of visited frames with their relevance scores and discards video–question pairs whose predicted answers do not match the ground truth.

**Stage II: Filtering and Verification.** After obtaining candidate keyframes from Stage I, we first discard frames with relevance scores below a threshold $\lambda$, leaving only those with stronger relevance. For each video–question pair, we

then test whether the selected keyframes alone are sufficient to answer the question. Specifically, the keyframe set and the question are fed into three independent VLMs (i.e., Qwen2.5-VL-72B (Bai et al., 2025b), LLaVA-OneVision-72B (Li et al., 2025a), and LLaVA-Video-72B (Zhang et al., 2025e)), and their predictions are compared with the ground-truth answer. Only instances for which all three models succeed using only the keyframes are retained. This cross-model verification ensures that the released dataset contains consistent, question-grounded keyframe annotations.

### 3.2. Dataset Statistics

Our two-stage pipeline produces 40,992 video–question pairs, forming the FrameOracle-41K dataset. Across video–question pairs, the median number of selected keyframes is five, the mean is around seven, and over 80% of instances require no more than 10 frames, while a small fraction of more complex cases require 30 or more frames. Figure 2 (left) categorizes all questions into 16 types following the taxonomy of LLaVA-Video-178K (Zhang et al., 2025e), covering a broad spectrum of reasoning skills such as description, localization, temporal understanding, and causal inference. Figure 2 (right) shows the per-type distribution of minimal keyframes. *Spatial* questions require the fewest frames (about 5.3 frames), while *Binary* questions need the most (around 13 frames). *Spatial* questions focus on static layouts within a scene, whereas *Binary* questions often ask whether an event occurs at any moment, requiring inspection of a broader temporal range. Some categories, such as *Camera Direction*, *Temporal*, and *Binary*, show high intra-class variability, with the number of required

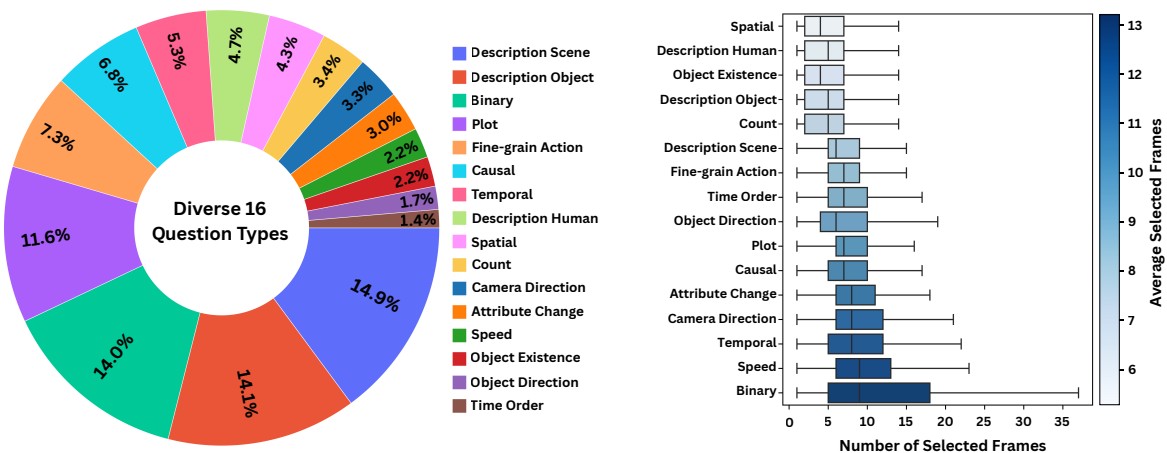

*Figure 2.* **FrameOracle-41K question-level statistics. Left:** Distribution of 16 question types across the dataset. **Right:** Per-type distributions of minimal sufficient keyframes.

frames varying widely across instances. This indicates that even within a single reasoning type, temporal complexity and evidence density can differ significantly, highlighting the heterogeneous nature of FrameOracle-41K and motivating adaptive frame selection. Complete dataset statistics, question type definitions, textual analyses, and an analysis of how the two-stage construction process affects dataset composition are provided in Appendix D.

## 4. FrameOracle

We introduce **FrameOracle**, a lightweight, plug-and-play frame selector that predicts which frames to select and how many to retain, conditioned on the query. By adaptively ranking and selecting frames from the candidate pool, FrameOracle provides the downstream VLM with a compact, highly relevant set of frames, enabling efficient video understanding. Processing all frames of a video, $V$, directly is computationally expensive. We therefore first apply uniform temporal sampling to extract a candidate set of $N$ frames, $V_C = \{f_1, \ldots, f_N\}$, reducing the input to a manageable size for FrameOracle (e.g., $N = 64$ or $N = 16$). Our goal is to learn FrameOracle as a *selection policy*, $\Pi_\theta$, parameterized by $\theta$, that operates on $V_C$ conditioned on a query $q$. Given $(V_C, q)$, $\Pi_\theta$ selects a compact subset of frames $V_S \subset V_C$ and dynamically determines its size $K = |V_S|$, unlike prior methods that fix the number of frames in advance. Formally, $\Pi_\theta$ maps $(V_C, q)$ to $V_S$, which is then passed to a downstream VLM, $\mathcal{M}$, producing an output $A = \mathcal{M}(V_S, q)$. FrameOracle is trained to maximize $\mathcal{M}$'s performance while keeping $K$ as small as possible, enabling efficient, query-conditioned video understanding.

### 4.1. Method

Each frame in $V_C$ is encoded with a visual encoder to produce a sequence of $N$ embeddings, while the query $q$ is tok-

enized and embedded using any text encoder. FrameOracle operates on these projected embeddings and is agnostic to the choice of tokenizer. The architecture, illustrated in Figure 3, consists of two main components: (1) a cross-modal fusion encoder that integrates video and query features, and (2) dual prediction heads that output frame relevance scores and the number of frames to retain.

**(1) Cross-Modal Fusion.** We fuse video and query embeddings using a cross-modal encoder to model interactions between the text query $q$ and the candidate frames $V_C$. Frame and text embeddings are first projected into a shared latent space via learnable linear layers, and then processed by a stack of Transformer encoder layers. We concatenate a learnable query token with the projected text and frame embeddings to form a single sequence $[\,k_{\text{query}}; \text{text}; \text{frames}\,]$, which is input to the encoder. This encoder is the sole module responsible for cross-modal interaction in FrameOracle and does not include any decoder. Its self-attention enables token-level reasoning, allowing each frame and text token to interact directly. Representing each frame with a single token ensures computational efficiency with minimal overhead for the downstream VLM.

**(2) Dual Prediction Heads.**

The fusion encoder outputs are fed into two specialized heads that implement the selection policy $\Pi_\theta$:

- **Rank Head:** Assigns a relevance score to each candidate frame $f_i \in V_C$ with respect to the query $q$. Processing the fused feature sequence, it outputs a scalar importance score $s_i$ for each frame, forming a vector $S = \{s_1, \ldots, s_N\}$. These scores determine which frames to select, i.e., *what to see* for the given query.

- **K Head:** Predicts the number of frames to retain, $K \le N$. It operates on globally aggregated features

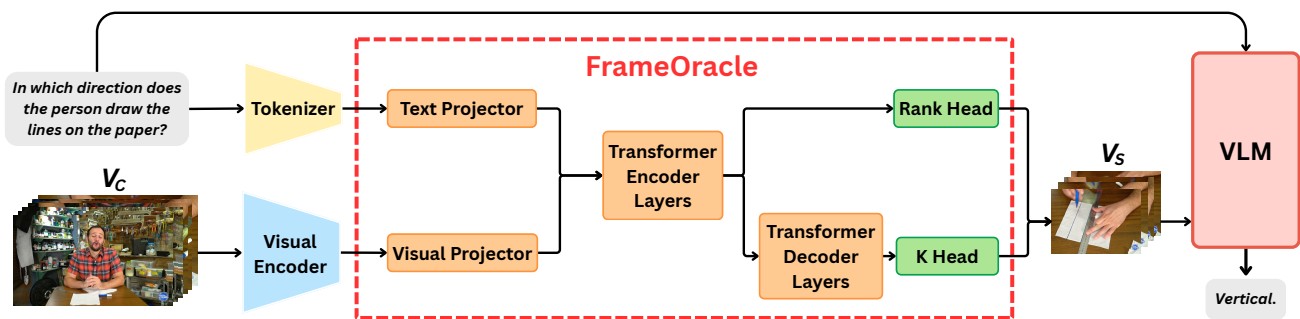

*Figure 3.* **Overview of the FrameOracle pipeline.** FrameOracle (dashed box) receives raw video frames and the textual prompt, and jointly predicts frame importance and the number of frames to keep. It outputs a compact keyframe subset, which is fed into the downstream VLM. $V_C$ denotes the pre-sampled frame collection, and $V_S$ denotes the subset selected by FrameOracle.

from the fusion encoder and outputs a probability distribution over possible values of $K$. A lightweight Transformer decoder is used only for frame-count prediction, separate from cross-modal fusion, implementing *how much to see*.

### 4.2. Training

Jointly learning which frames to select and how many to retain is challenging, as optimizing frame ranking and the predicted subset size simultaneously can destabilize training. To address this, we adopt a staged training strategy that gradually refines the selection policy $\Pi_\theta$: first learning robust frame and text representations, then frame ranking, and finally predicting the number of frames to keep. This approach enables effective reasoning over pre-sampled frame sets (e.g., 16 or 64 frames). Training leverages four widely used public VideoQA datasets covering clips from roughly 10 seconds to 15 minutes (details in Appendix A).

**Stage 1: Text-Visual Alignment.** This stage trains Frame-Oracle to produce robust cross-modal representations by aligning the textual query with the candidate frame set $V_C$. We use the pre-trained SigLIP model (Zhai et al., 2023) as a teacher, treating its similarity scores for each query-frame pair as target relevance signals. The frame and text feature projectors, along with the cross-modal Transformer encoder, are trained using a RankNet loss (Burges et al., 2005) to match the relative ordering of SigLIP similarities. Concretely, for frames $i$ and $j$ with SigLIP scores $s_i$ and $s_j$, and predicted scores $y_i$ and $y_j$, we define the pairwise label $t_{ij} = \text{sign}(s_i - s_j)$. The loss is

$$\mathcal{L}_{\text{RankNet}} = \sum_{i<j} \log\Big(1 + \exp\big(-t_{ij}\,(y_i - y_j)\big)\Big), \quad (1)$$

where $t_{ij} = 0$ (ties) do not contribute to gradients. The K Head is frozen during this stage, ensuring that learning focuses solely on frame ranking.

**Stage 2: Rank Head Optimization.** In this stage, the Rank Head is trained to identify the most salient frames in the candidate set $V_C$. Unlike Stage 1, which uses frame-level SigLIP scores independently, we leverage the downstream VLM's loss to capture frame importance in context, including temporal dependencies. For each frame $f_i \in V_C$, we adopt a leave-one-out (LOO) approach: $f_i$ is removed from the input set, and the resulting change in the VLM's loss indicates its importance. These LOO-based scores serve as soft targets, and the Rank Head is trained with a RankNet loss to predict them. The Transformer encoder and feature projectors are fine-tuned at a smaller learning rate for stability, while the K Head remains frozen.

**Stage 3: K Head Optimization.** In this stage, the K Head learns to predict the number of frames to select, $K$. The Rank Head is frozen, while the Transformer encoder and feature projectors are fine-tuned with a very small learning rate for slight adaptation. For each sample, we evaluate the downstream VLM on subsets of top-$k$ frames from $V_C$, ranked by the frozen Rank Head, for $k \in \{1, \ldots, N\}$. The target number of frames is chosen as

$$k^* = \arg\min_{k \in \{1,\ldots,N\}} \Big(\text{zscore}(\mathcal{L}_{\text{task}}(k)) + \lambda_k\, k\Big), \quad (2)$$

where the linear penalty $\lambda_k k$ balances accuracy and frame cost. The K Head predicts a categorical distribution $p_\theta(k)$ over possible $k$ values and is trained with a combination of the Expected Value Objective and a classification loss:

$$\mathcal{L}_K = (1-\alpha)\,\mathcal{L}_{\text{evo}} + \alpha\,\mathcal{L}_{\text{class}}, \quad (3)$$

where

$$\mathcal{L}_{\text{evo}} = \text{SmoothL1}\left(\sum_{k=1}^{N} k\, p_\theta(k),\ k^*\right)$$

regresses the expected frame count to $k^*$, and $\mathcal{L}_{\text{class}}$ aligns $p_\theta$ with a Gaussian-shaped soft target centered at $k^*$.

**Stage 4: Supervised Fine-tuning with Keyframe Annotations.** In the final stage, FrameOracle is fine-tuned on FrameOracle-41K, which provides verified annotations for

both keyframe indices and the number of frames, $K$. The Rank Head is trained to align its predictions with the annotated keyframes, while the K Head is jointly trained to predict the annotated $K$ values. Unlike the previous stages, which rely on weak or proxy signals, this direct supervision consolidates the selection policy, ensuring that FrameOracle accurately identifies both *what to see* and *how much to see* for each query.

## 5. Experiments

### 5.1. Experiment Settings

**Implementation Details.** We train two FrameOracle versions, taking 16 or 64 uniformly sampled frames as input. Both use DINOv2 (Oquab et al., 2024) as the visual encoder and Qwen2.5-VL as the text tokenizer. Training follows the staged curriculum described in Section 4.2. For supervision in Stages 2 and 3, we adopt Qwen2.5-VL-3B as the backbone VLM. For the 64-frame selector, the predicted $K$ is capped at 16 during Stage 3 to ensure comparability with experiments. All training runs use 8×H100 GPUs. Full hyperparameter details, including learning rates, batch sizes, and per-stage training times, are provided in Appendix A.

**Benchmarks.** We evaluate FrameOracle on six widely adopted video benchmarks, which can be divided into long-video and short-video understanding tasks. For long-video understanding, we include EgoSchema (Mangalam et al., 2023) (all videos 3 min), LongVideoBench (Wu et al., 2024) (average ~12 min), MLVU (Zhou et al., 2025) (average ~12 min), and Video-MME (Fu et al., 2025) (average ~17 min), all of which require reasoning over extended temporal contexts **ranging from minutes to hours**. These datasets emphasize challenges such as cross-event reasoning, global consistency, and temporal grounding across lengthy sequences. For short-video understanding, we evaluate on NExTQA (Xiao et al., 2021) and Perception (Pătrăucean et al., 2023), which contain clips typically under one minute and focus on fine-grained event recognition, local temporal relations, and reasoning within concise videos. Evaluation follows the LMMs-Eval library (Zhang et al., 2025b), and we report accuracy across all benchmarks.

### 5.2. Results and Analysis

**Comparison with State-of-the-Art and FrameOracle-Enhanced Models.** Table 1 compares FrameOracle under two evaluation settings: (1) standalone performance of four state-of-the-art (SOTA) VLM baselines, and (2) FrameOracle integrated as a frame selector with five diverse VLM backbones. For each backbone integrated with FrameOracle, we evaluate two configurations, using candidate pools of 16 and 64 uniformly sampled frames. The Qwen-VL series internally merges every two adjacent frames into a single

visual representation. To ensure a fair comparison with models that process raw frames, we report Qwen baselines with 32 and 128 input frames.

With a 16-frame candidate pool, FrameOracle preserves accuracy across all benchmarks while reducing the downstream VLM's frame processing by approximately 35%. When scaling the candidate pool to 64 frames, FrameOracle benefits from denser temporal coverage and adaptively selects more informative subsets, consistently improving accuracy over the corresponding baselines by an average of 1.5% while still reducing the effective frame count by about 15%. These results indicate that larger candidate pools allow FrameOracle to better exploit temporal redundancy, yielding stronger accuracy–efficiency trade-offs.

This scaling behavior is particularly evident in long-video settings. For example, starting from 128 candidate frames, Qwen3-VL-8B equipped with FrameOracle outperforms Video-XL-7B, which processes 256 frames, and even Video-XL-2-8B, which operates on approximately 10k frames, on LongVideoBench (average video length ~12 min) and Video-MME (average video length ~17 min). Since FrameOracle is trained independently and applied in a fully plug-and-play manner, without co-training or backbone-specific adaptation, these results demonstrate its strong generalization across model architectures.

**RQ 1: Does giving a VLM more frames consistently improve its performance?**

One might expect that providing more frames improves performance by offering additional visual evidence. However, Table 1 shows that using more frames often fails to help and can even reduce accuracy. This observation aligns with recent findings that long-video reasoning is inherently sparse, with only a small subset of frames being truly relevant (Park et al., 2026). Additional frames primarily introduce redundancy and noise, which can dilute cross-modal attention and lead to diminishing returns (Li et al., 2023).

In contrast, when FrameOracle selects a smaller but more informative subset of frames, performance improves, particularly on open-ended benchmarks. For example, on LLaVA-OneVision-7B, FrameOracle reduces the input from 16 frames to about 10 frames on average, while improving NExTQA performance (OE_val: 14.6 → 16.1, OE_test: 16.7 → 17.8) and EgoSchema (60.8 → 62.4). Qualitative examples in Appendix I further illustrate that FrameOracle isolates the critical visual evidence, yielding correct answers where naive higher-frame sampling fails.

Crucially, these improvements do not arise from simply reducing the number of input frames. As shown in our ablation study (Appendix F.2), uniformly sampling 10 frames from the same 16-frame input substantially degrades performance (49.8% → 46.3% on LLaVA-OneVision-7B). In contrast,

| Model | Frames | NExTQA | | | Perception | LVB | Video-MME | EgoSchema | MLVU | Avg. |
|---|---|---|---|---|---|---|---|---|---|---|
| | | OE_val | OE_test | MC | | | | | | |
| *(1) State-of-the-Art Models* | | | | | | | | | | |
| VideoChat2-7B (Li et al., 2024) | 16 | - | - | - | - | 39.3 | 39.5 | 63.6 | 44.5 | - |
| VideoLLaMA2-7B (Cheng et al., 2024) | 16 | - | - | 45.4 | 54.9 | 53.1 | 47.9 | 53.1 | - | - |
| Video-XL-7B (Shu et al., 2025) | 256 | - | - | - | - | 49.5 | 64.0 | - | 64.9 | - |
| Video-XL-2-8B (Qin et al., 2025) | ~10k | - | - | - | - | 61.0 | 66.6 | - | 74.8 | - |
| *(2) FrameOracle on Different Baselines* | | | | | | | | | | |
| Qwen2.5-VL-3B (Bai et al., 2025b) | 32 | 25.1 | 29.6 | 75.4 | 65.9 | 54.1 | 58.4 | 53.4 | 59.4 | 52.7 |
| + **FrameOracle** | 32→20.9 | 25.6 | 30.5 | 74.8 | 66.7 | 54.3 | 58.5 | 53.8 | 58.4 | 52.8 |
| + **FrameOracle** | 128→27.8 | **26.0** | **31.7** | **76.1** | **67.8** | **54.8** | **59.7** | **54.5** | **61.6** | **54.0** |
| LLaVA-OneVision-7B (Li et al., 2025a) | 16 | 14.6 | 16.7 | 78.2 | 56.4 | 55.0 | 56.1 | 60.8 | 60.9 | 49.8 |
| + **FrameOracle** | 16→10.4 | 16.1 | 17.8 | 77.6 | 56.5 | 55.5 | 56.0 | 62.4 | 60.2 | 50.3 |
| + **FrameOracle** | 64→13.9 | **16.5** | **19.0** | **78.5**§ | **56.9** | **56.5** | **58.1** | **63.4** | **63.7** | **51.6** |
| LLaVA-Video-7B (Zhang et al., 2025e) | 16 | 27.3 | 32.4 | 81.0 | 64.3 | 55.8 | 59.8 | 54.2 | 61.7 | 54.6 |
| + **FrameOracle** | 16→10.4 | 27.8 | 33.0 | 80.4 | 64.7 | 56.3 | 59.6 | 54.6 | 60.8 | 54.7 |
| + **FrameOracle** | 64→13.9 | **28.8** | **33.9** | **81.6** | **65.1** | **57.8** | **61.6** | **55.2** | **64.3** | **56.0** |
| VideoLLaMA3-7B (Zhang et al., 2025a) | 16 | 27.8 | 32.3 | **82.3** | 72.3 | 56.1 | 61.2 | 61.4 | 50.9 | 55.5 |
| + **FrameOracle** | 16→10.4 | 28.3 | 32.9 | 81.2 | 72.0 | 56.0 | 61.4 | 61.8 | 52.8 | 55.8 |
| + **FrameOracle** | 64→13.9 | **28.9** | **33.6** | 82.0§ | **72.8** | **56.9** | **61.8** | **62.4** | **54.1** | **56.6** |
| Qwen3-VL-8B (Bai et al., 2025a) | 32 | 26.0 | 31.1 | 76.6 | 67.5 | 63.3 | 66.9 | 70.8 | 63.6 | 58.2 |
| + **FrameOracle** | 32→20.9 | 26.6 | 32.3 | 76.1 | 68.2 | 64.0 | 67.3 | 71.4 | 62.9 | 58.6 |
| + **FrameOracle** | 128→27.8 | **28.1** | **33.8** | **77.3** | **69.0** | **65.2** | **69.1** | **72.3** | **66.3** | **60.1** |

*Table 1.* **FrameOracle vs. SOTA VLMs.** "Frames" shows $M \to \bar{K}$: FrameOracle starts from $M$ uniformly sampled frames and reduces to an average of $\bar{K}$ frames. Highlighted rows correspond to larger frame inputs. LVB = LongVideoBench validation set. For each backbone model, statistical significance is evaluated for the best-performing FrameOracle variant. 95% confidence intervals (CIs) are computed via paired bootstrap resampling over questions (10,000 iterations, percentile method). Improvements are considered statistically significant at $\alpha = 0.05$ if the CI excludes zero. § denotes non-significant changes (95% CI includes zero).

FrameOracle improves accuracy to 50.3% with the same frame budget. This demonstrates that the gains stem from selecting semantically relevant frames, rather than merely reducing the number of visual tokens.

## RQ 2: How does FrameOracle compare with existing SOTA methods for keyframe selection?

We compare FrameOracle with existing keyframe selection methods, including (1) jointly trained approaches and (2) plug-and-play selectors applied to open-source VLMs (Table 2). Because jointly trained methods are not directly transferable, we focus on plug-and-play comparisons under a unified 8-frame evaluation setting. For fairness, we disable FrameOracle's K Head and use only the Rank Head to select the top-8 frames from uniformly sampled candidates.

FrameOracle performs strongly across multiple backbones and benchmarks. On LLaVA-OneVision-7B and LLaVA-Video-7B, it consistently outperforms existing plug-and-play selectors such as Frame-Voyager (Yu et al., 2025), BOLT (Liu et al., 2025), and KFC (Fang et al., 2026) on NExTQA, LongVideoBench, Video-MME, and EgoSchema. On the Qwen2.5-VL-7B backbone, FrameOracle also compares favorably against recent strong baselines, outperforming ViaRL (Xu et al., 2025) and matching or exceeding K-frames (Yao et al., 2025), while operating with a smaller candidate pool (128 vs. 256 frames).

On MLVU, FrameOracle improves over the base VLMs but does not consistently outperform all SOTA methods when using a 64-frame candidate pool. MLVU includes very long videos, some exceeding 2 hours, emphasizing broad temporal coverage. Several SOTA methods address this challenge by operating with substantially higher temporal coverage, using 128 or 256 frames, or sampling at fixed rates such as 1 FPS across the entire video. To explore how FrameOracle behaves under increased coverage, we evaluate a variant that divides the video into multiple temporal segments and applies FrameOracle independently within each segment, effectively increasing coverage while keeping the final frame budget fixed. Results in Appendix F.3 show that this strategy leads to improved performance on MLVU.

Overall, these results show that the Rank Head alone can reliably prioritize informative frames and achieve state-of-the-art performance across most benchmarks, even without adaptive frame-count prediction.

## RQ 3: How effectively can FrameOracle reduce computation while maintaining accuracy?
We take LLaVA-Video-7B with 16 input frames as the baseline and report per-GPU, per-sample averages. FrameOracle reduces the input from 16 to 10.4 frames, cutting the VLM cost from 184.38 to 109.11 TFLOPs and the end-to-end total to 110.98 TFLOPs ($-39.8\%$). It also lowers latency from 0.615 to 0.363 seconds ($-41.0\%$) and reduces tokens from 11,644.0 to 7,581.6, while maintaining accuracy. With a larger candidate pool, FrameOracle reduces 64 frames to 13.9, im-

| Model | Frames | NExTQA | LVB | Video-MME | EgoSchema | MLVU |
|---|---|---|---|---|---|---|
| *(1) Jointly Trained Keyframe Selection Methods* | | | | | | |
| SeViLA (Yu et al., 2023) | 8 | 63.6 | - | - | 25.7 | - |
| VideoAgent (Wang et al., 2024) | 8.4 | 71.3 | - | - | 60.2 | - |
| FFS (Buch et al., 2025) | 8.6 | 66.7 | - | - | - | - |
| AKS (Tang et al., 2025) | 64 | - | 62.7 | 65.3 | - | - |
| *(2) Plug-and-Play Keyframe Selection Methods* | | | | | | |
| LLaVA-OneVision-7B (Li et al., 2025a) | 8 | 77.4 | 54.3 | 53.8 | 62.0 | 58.4 |
| + Frame-Voyager (Yu et al., 2025) | 128→8 | 73.9 | - | **57.5** | - | **65.6** |
| + BOLT (Liu et al., 2025) | 1fps→8 | 77.4 | 55.6 | 56.1 | 62.2 | 63.4 |
| + KFC (Fang et al., 2026) | 1fps→8 | - | 55.6 | 55.4 | - | 65.0 |
| **+ FrameOracle** | 64→8 | **77.8** | **56.0** | **57.5** | **62.8** | 62.9 |
| LLaVA-Video-7B (Zhang et al., 2025e) | 8 | 75.6 | 54.2 | 55.9 | 51.8 | 60.5 |
| + BOLT (Liu et al., 2025) | 1fps→8 | - | - | 58.6 | - | - |
| + KFC (Fang et al., 2026) | 1fps→8 | - | 56.5 | 57.6 | - | **66.9** |
| **+ FrameOracle** | 64→8 | **76.5** | **56.9** | **58.9** | **53.0** | 63.4 |
| Qwen2.5-VL-7B (Bai et al., 2025b) | 8 | 69.2 | 53.6 | 54.1 | 56.8 | 54.5 |
| + ViaRL (Xu et al., 2025) | 128→8 | - | - | 57.3 | - | 58.2 |
| + K-frames (Yao et al., 2025) | 256→8 | - | 57.7 | **57.4** | - | **60.4** |
| **+ FrameOracle** | 128→8 | **71.7** | **59.1** | 57.4 | **59.5** | 59.6 |

*Table 2.* **FrameOracle vs. SOTA keyframe selection methods.** NExTQA reports MCQ. Methods using more frames or larger LLMs are shown in gray. LVB = LongVideoBench validation set. For fair comparison under a fixed budget, we disable FrameOracle's K Head and use only the Rank Head to select the top-8 frames from uniformly sampled candidates. For all backbone models, FrameOracle achieves statistically significant improvements over the corresponding baselines (paired bootstrap, $\alpha = 0.05$).

| Model | Frames | TFLOPs ↓ | | | | Latency (s) ↓ | Visual Tokens ↓ | Avg. Acc. ↑ |
|---|---|---|---|---|---|---|---|---|
| | | DINOv2 | FrameOracle | VLM | Total | | | |
| LLaVA-Video-7B | 16 | – | – | 184.38 | 184.38 | 0.615 | 11,644.0 | 54.6 |
| LLaVA-Video-7B | 32 | – | – | 405.64 | 405.64 | 1.140 | 23,290.0 | 56.2 |
| LLaVA-Video-7B | 64 | – | – | 792.83 | 792.83 | 2.622 | 46,584.0 | **56.6** |
| + FrameOracle | 16→10.4 | 1.87 | $2.6 \times 10^{-4}$ | **109.11** | **110.98** | **0.363** | **7,581.6** | 54.7 |
| + FrameOracle | 64→13.9 | 7.58 | $1.0 \times 10^{-3}$ | 160.09 | 167.67 | 0.556 | 10,133.1 | 56.0 |

*Table 3.* **Comparison of FLOPs, latency, visual tokens, and accuracy.** The values of the computational cost are reported as per-GPU, per-sample averages.

proving accuracy by +1.4 while still lowering total compute to 167.67 TFLOPs (−9.1%), tokens to 10,133.1 (−13.0%), and latency to 0.556 seconds (−9.6%). Compared with simply increasing the raw frame budget, FrameOracle also provides a more favorable efficiency-accuracy trade-off. The raw 32-frame baseline achieves only +0.2 higher accuracy than the 64→13.9 FrameOracle setting, but requires more than 2× higher total latency. The raw 64-frame baseline further improves accuracy by only +0.6, while increasing compute by roughly 5×. These results highlight two complementary trends: smaller candidate pools yield larger efficiency gains without harming accuracy, while larger pools enable more significant accuracy improvements with moderate compute savings. In contrast, simply increasing the raw frame budget yields only small gains in accuracy at a much higher computational cost. Notably, roughly 90% of the total computation comes from the backbone VLM, with FrameOracle contributing only about 10% of the cost. As shown in Table 3, moving from the 16 → 10.4 to the 64 →

13.9 setting increases total FLOPs almost entirely due to the VLM. While the exact trade-offs vary, the observed efficiency–accuracy patterns are likely to extend across models of different sizes and architectures, as they primarily stem from the relative cost distribution between the lightweight selector and the backbone VLM.

**RQ 4: How does each training stage contribute?** We conduct ablations over the four training stages using Qwen2.5-VL-3B as the backbone VLM (Table 4). The baseline (first row) randomly selects 16 frames from 32 uniformly sampled candidates. Stage 1 (text–visual alignment) alone underperforms this baseline, as it focuses solely on learning cross-modal representations and is not designed to directly optimize downstream task performance. Nevertheless, it provides a necessary initialization for subsequent stages. Stage 2 (frame ranking) and Stage 3 (frame-count prediction) each independently improve performance when applied in isolation. Finally, incorporating Stage 4, which fine-tunes FrameOracle on FrameOracle-41K, further im-

| Model | Frames | NExTQA | | | Perception | LVB | Video-MME | EgoSchema | MLVU |
|---|---|---|---|---|---|---|---|---|---|
| | | OE_val | OE_test | MC | | | | | |
| Qwen2.5-VL-3B | 32→16 | 23.4 | 29.1 | 71.9 | 65.0 | 52.9 | 54.8 | 50.2 | 56.7 |
| + Stage 1 | 32→16 | 24.7 | 29.2 | 72.4 | 60.3 | 49.8 | 52.4 | 48.2 | 51.3 |
| + Stage 2 | 32→16 | 24.8 | 29.5 | 73.0 | 64.7 | 51.9 | 55.7 | 52.2 | 54.8 |
| + Stage 3 | 32→21.8 | 25.1 | 30.0 | 74.1 | 66.0 | 53.7 | **59.4** | 53.6 | 57.6 |
| + Stage 4 | 32→20.9 | **25.6** | **30.5** | **74.8** | **66.7** | **54.3** | 58.5 | **53.8** | **58.4** |

*Table 4.* **Four-stage training of FrameOracle on Qwen2.5-VL-3B.** Stages are added progressively to assess their impact. The baseline (first row) randomly selects 16 of 32 frames. **Bold** numbers indicate best performance. LVB = LongVideoBench validation set.

| Training Strategy | Frames | NExTQA | | | Perception | LVB | Video-MME | EgoSchema | MLVU |
|---|---|---|---|---|---|---|---|---|---|
| | | OE_val | OE_test | MC | | | | | |
| Stage 2+3 (joint) | 32→16 | 24.0 | 27.8 | 72.2 | 63.8 | 50.1 | 54.3 | 49.8 | 52.2 |
| Stage 2 alone | 32→16 | **24.8** | **29.5** | **73.0** | **64.7** | **51.9** | **55.7** | **52.2** | **54.8** |

*Table 5.* **Joint Training vs. Staged Training.** Comparison of downstream performance with a fixed 16-frame budget.

proves accuracy on most benchmarks while reducing the average number of selected frames.

**RQ 5: Can we jointly train Stage 2 and 3?** We further investigate whether Stage 2 (Rank optimization) and Stage 3 (K optimization) could be simplified into a single joint training phase. We train a variant where both heads are optimized simultaneously. We observe severe optimization instability under the joint setting. Specifically, the K Head collapses, predicting near-maximum frames (i.e., failing to perform meaningful frame reduction), and the Rank Head becomes unstable, with Kendall-$\tau$ fluctuating between -0.4 and +0.6. This instability arises because the two objectives interfere during joint optimization: immature Rank predictions generate noisy top-K subsets that destabilize K-Head learning, and the resulting unstable K outputs further corrupt Rank learning, creating a feedback loop that prevents either head from converging. To quantify the impact, we measure the ranking consistency (Kendall-$\tau$) against ground truth and evaluate downstream performance under a fixed 16-frame budget. As shown in Table 5, the jointly trained model achieves a Kendall-$\tau$ of only **0.2313** (vs. 0.5367 for Stage 2 alone) and consistently underperforms the staged baseline across all benchmarks. These results confirm that decoupling the ranking and budgeting objectives via a curriculum is essential for stable optimization.

**RQ 6: Can we skip the intermediate stages?** To assess whether the model could learn solely from Stage 4's strong supervision (FrameOracle-41K), we perform an ablation that compares the full pipeline against a simplified version using only Stage 1 and Stage 4, evaluating two settings: (1) fixed $K$ and (2) adaptive $K$ prediction. As Table 6 shows, skipping intermediate stages reduces performance. Stage 1+4 improves over Stage 1 alone but still falls short of the Stage 1+2 baseline. Notably, enabling the K Head without Stage 3 calibration (Adaptive K) reduces accuracy to 46.9%

| Setting | Frames | Avg. Acc. |
|---|---|---|
| Qwen2.5-VL-3B (Baseline) | 32 | 50.5 |
| + Stage 1 | 32 → 16 | 48.5 |
| + Stage 1+4 (Fixed K) | 32 → 16 | 49.6 |
| + Stage 1+2 | 32 → 16 | 50.8 |
| + Stage 1+4 (Adaptive K) | 32 → 13.4 | 46.9 |
| **+ Full (Stage 1–4)** | **32 → 20.9** | **52.8** |

*Table 6.* **Ablation of training stages.** Results are reported on Qwen2.5-VL-3B with 32 candidate frames. Average accuracy is calculated across all benchmarks listed in Table 1.

and leads to overfitting on FrameOracle-41K. These results confirm that weak supervision in Stages 2 and 3 is crucial for learning generalized ranking and frame-budget policies prior to refinement using reference keyframe annotations.

## 6. Conclusion

We propose FrameOracle, a lightweight, plug-and-play frame selector that adaptively determines both which frames to select and how many are needed from a given candidate pool. To support training, we introduce FrameOracle-41K, a large-scale VideoQA dataset with 40,992 examples and the first to provide question-conditioned keyframe annotations that specify the minimal frames required to answer each query. Extensive experiments demonstrate that FrameOracle consistently improves a wide range of VLM backbones without co-training, while reducing FLOPs, inference latency, and visual token usage, and outperforming state-of-the-art keyframe selection methods. While FrameOracle-41K provides high-quality, large-scale supervision for adaptive frame selection, expanding the dataset with more diverse videos and question types could further strengthen frame-level supervision and enable training FrameOracle primarily from strong keyframe annotations, potentially simplifying the current multi-stage training pipeline.

## Impact Statement

This paper presents work whose goal is to advance the field of machine Learning. There are many potential societal consequences of our work, none of which we feel must be specifically highlighted here.

## Acknowledgments

This research was partially supported by the National Eye Institute (NEI) of the National Institutes of Health (NIH) under award number R01EY034562. The content is solely the responsibility of the authors and does not necessarily represent the official views of the NIH.

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

# A. Full Implementation Details

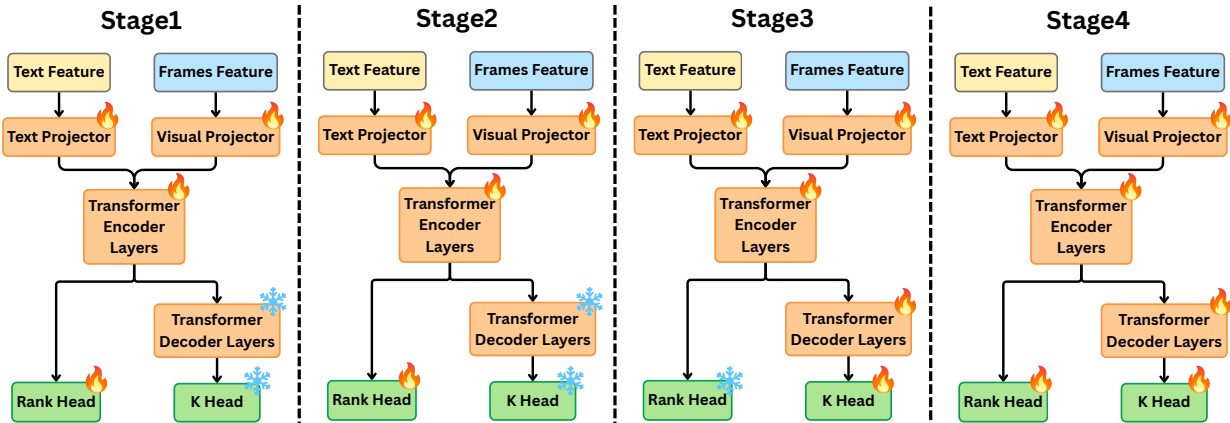

*Figure 4.* **Four-stage training strategy of FrameOracle.** The model is progressively optimized from weak to strong supervision, culminating in supervised fine-tuning with FrameOracle-41K annotations. Fire icons indicate trainable modules, while snowflake icons denote frozen ones.

| Task | Dataset | Amount |
|------|---------|--------|
| Stage1 | LLaVA-Video-178K (Zhang et al., 2025e), ShareGPT4o-Video (Chen et al., 2024b), Video-ChatGPT (Maaz et al., 2024) | 300K |
| Stage2 | LLaVA-Video-178K (Zhang et al., 2025e), LLaVA-Hound (Zhang et al., 2025c), Video-ChatGPT (Maaz et al., 2024) | 300K |
| Stage3 | LLaVA-Video-178K (Zhang et al., 2025e), LLaVA-Hound (Zhang et al., 2025c), Video-ChatGPT (Maaz et al., 2024) | 300K |
| Stage4 | **FrameOracle-41K (Our Dataset)** | 40K |

*Table 7.* **Datasets used in FrameOracle training.** Stage 4 leverages FrameOracle-41K.

**Training strategy illustration.** Figure 4 presents a schematic of our four-stage curriculum, highlighting trainable modules (fire) and frozen modules (snowflake) at each stage.

**Hardware and input budgets.** All training is conducted on $8 \times$H100 GPUs. We train two FrameOracle variants: one with 16 uniformly sampled candidate frames and another with 64. A cosine learning rate scheduler with the AdamW optimizer is used across all stages.

**Datasets used in staged training.** FrameOracle is optimized using a four-stage curriculum with progressively stronger supervision. Stages 1-3 rely on large-scale video–language corpora, while Stage 4 leverages our FrameOracle-41K dataset. Table 7 summarizes the dataset composition for each stage.

**Stage 1: Cross-modal alignment.** K Head is frozen while the feature projectors and cross-modal Transformer encoder are trained jointly, both optimized with a learning rate of $1 \times 10^{-4}$. The 16-frame selector uses a batch size of 16 and trains for approximately 48 hours, whereas the 64-frame version uses a smaller batch size of 2 and completes in about 91 hours.

**Stage 2: Rank Head optimization.** Rank Head is trained while the K Head remains frozen. The Rank Head uses a learning rate of $1 \times 10^{-4}$, and the feature projectors and Transformer encoder are fine-tuned with a smaller learning rate of $1 \times 10^{-5}$. The 16-frame selector uses a batch size of 16 and trains for approximately 40 hours, whereas the 64-frame variant uses the same batch size and takes about 52 hours.

**Stage 3: K Head optimization.** K Head is the primary trainable module, optimized with a learning rate of $1 \times 10^{-4}$. The feature projectors and Transformer encoder are lightly updated with a learning rate of $1 \times 10^{-7}$, while the Rank Head remains frozen. We set $\lambda_k = 0.0105$ to balance accuracy and efficiency. The 16-frame selector uses a batch size of 16 and trains for approximately 35 hours, whereas the 64-frame variant uses the same batch size and takes about 60 hours.

**Stage 4: Supervised fine-tuning on FrameOracle-41K.** Rank Head and K Head are trained jointly with a learning rate of $5 \times 10^{-5}$, while the feature projectors and Transformer encoder are fine-tuned with $1 \times 10^{-5}$. The 16-frame selector is trained with a batch size of 8 for approximately 12 hours, and the 64-frame version uses the same batch size and trains for about 18 hours.

## B. FrameOracle-41K Data Format

We release the FrameOracle-41K dataset in JSON format, with each entry corresponding to a single video–question pair. Each entry includes the instance identifier, question–answer pair, paths to the associated video and extracted keyframes, video duration, and number of selected frames. Below, we provide an example JSON entry to illustrate the dataset's structure.

```
{
  "id": 30,
  "question": "What folding technique is demonstrated first in the video?",
  "ground_truth_answer": "The 'SHIKAKU NO GI' (Square Fold) technique is demonstrated
      first.",
  "video": "/srv/nfs/video_data/video/ytb_8yhoV5C3bT8.mp4",
  "keyframes_dir": "/srv/nfs/video_data/extracted_frames/ytb_8yhoV5C3bT8",
  "duration": 126.893,
  "num_selected_frames": 8
}
```

## C. Prompts for Data Generation

> **Prompt Template for Stage I: Initial Frame Analysis**
>
> You are analyzing a video that is {duration_seconds} seconds long. The video has been uniformly sampled into 64 frames, indexed from 0 (start) to 63 (end).
>
> Analyze these {len(initial_indices)} initial frames (indices: {initial_indices}) to answer: "{question}". Provide a short caption for each frame, a relevance score (INTEGER 1-5), your confidence (high/medium/low), and your answer attempt.
>
> Respond in JSON: {{"frame_analysis": [{{"index": int, "caption": "str", "relevance": int}}], "confidence": "str", "answer_attempt": "str", "reasoning": "str"}}
>
> **IMPORTANT GUIDELINES:**
>
> - Relevance combines BOTH
> (a) how well the frame's TEMPORAL POSITION matches the question mentioned, and
> (b) how much the visible CONTENT answers the question. A high score (4-5) requires strong evidence on both axes.
> - You may use "high" confidence early ONLY IF: You have seen explicit, definitive evidence that unquestionably answers the question (e.g., clearly visible target object/person/action).
> - Before setting "high" confidence, explicitly mention in your reasoning:
> (a) Why current evidence is sufficient.
> (b) Why additional unseen frames are unlikely to alter your conclusion.
> - If there's any reasonable scenario where unseen frames could alter your answer, you must explicitly acknowledge that and keep your confidence at "medium".
>
> **Follow these instructions strictly.**

---

**Prompt Template for Stage II: Deep-dive Analysis and Refinement**

You are analyzing a video that is {duration_seconds} seconds long. The video has been uniformly sampled into 64 frames, indexed from 0 (start) to 63 (end).

Current context on question "{question}":

Current context in buffer "{buffer}".

Now analyze these {len(indices)} new frames (indices: {[int(idx) for idx in indices]}) from the gap ({start_idx}, {end_idx}).

**Tasks:**
- Provide a caption, relevance score (INTEGER 1-5) for each NEW frame, your UPDATED confidence, answer, and reasoning.
- If the new evidence changes your view of any PREVIOUS frame listed above, list the updated scores under "revised_prev_scores" (index, new relevance 1-5).

Respond in JSON: {{"new_frame_analysis": [{{"index": int, "caption": "str", "relevance": int}}], "revised_prev_scores": [{{"index": int, "relevance": int}}], "confidence": "str", "answer_attempt": "str", "reasoning": "str"}}

---

## D. Additional Dataset Statistics

To complement the main dataset description, we provide additional statistics that illustrate the composition, visual properties, and textual characteristics of FrameOracle-41K. We include both general dataset statistics and a sanity check on the stability of dataset composition under the two-stage construction process. Table 8 lists the 16 question types used in the dataset along with their corresponding definitions.

Figure 5 visualizes several key quantitative aspects of the dataset. Figure 5a shows the distribution of video durations in FrameOracle-41K, indicating that most videos are between two and three minutes long. This provides sufficient temporal

| Question type | Definition |
|---|---|
| Temporal | Designed to assess reasoning about temporal relationships between actions or events. Questions involve previous, present, or next actions. |
| Spatial | Tests ability to perceive spatial relationships between observed instances in a video scene. |
| Causal | Focuses on explaining actions or events and determining intentions, causes, or consequences. |
| Description Scene | Assesses ability to describe the major scene of the video, such as where it takes place and the overall environment. |
| Description Human | Involves describing actions or attributes of people, such as their activities and appearances. |
| Description Object | Assesses ability to describe attributes of objects, including appearance and function. |
| Count | Tests ability to count instances of objects, people, or actions, and to distinguish between old and new elements in a scene. |
| Binary | Involves yes/no questions related to the video content. |
| Fine-Grained Action | Creates questions that challenge comprehension of subtle or detailed actions. |
| Plot | Challenges ability to interpret the narrative or plot in the video. |
| Object Existence | Assesses reasoning with introduced non-existent activities while keeping physical scene details unchanged. |
| Time Order | Challenges recognition of the temporal sequence of activities in videos. |
| Object Direction | Emphasizes perception of object movement direction. |
| Camera Direction | Focuses on the direction of camera movement. |
| Speed | Delves into discerning variations in motion speed, including absolute and relative differences. |
| Attribute Change | Centers on how object or scene attributes change over time, such as size, shape, color, or other properties. |

*Table 8.* Question types and their corresponding definitions in FrameOracle-41K.

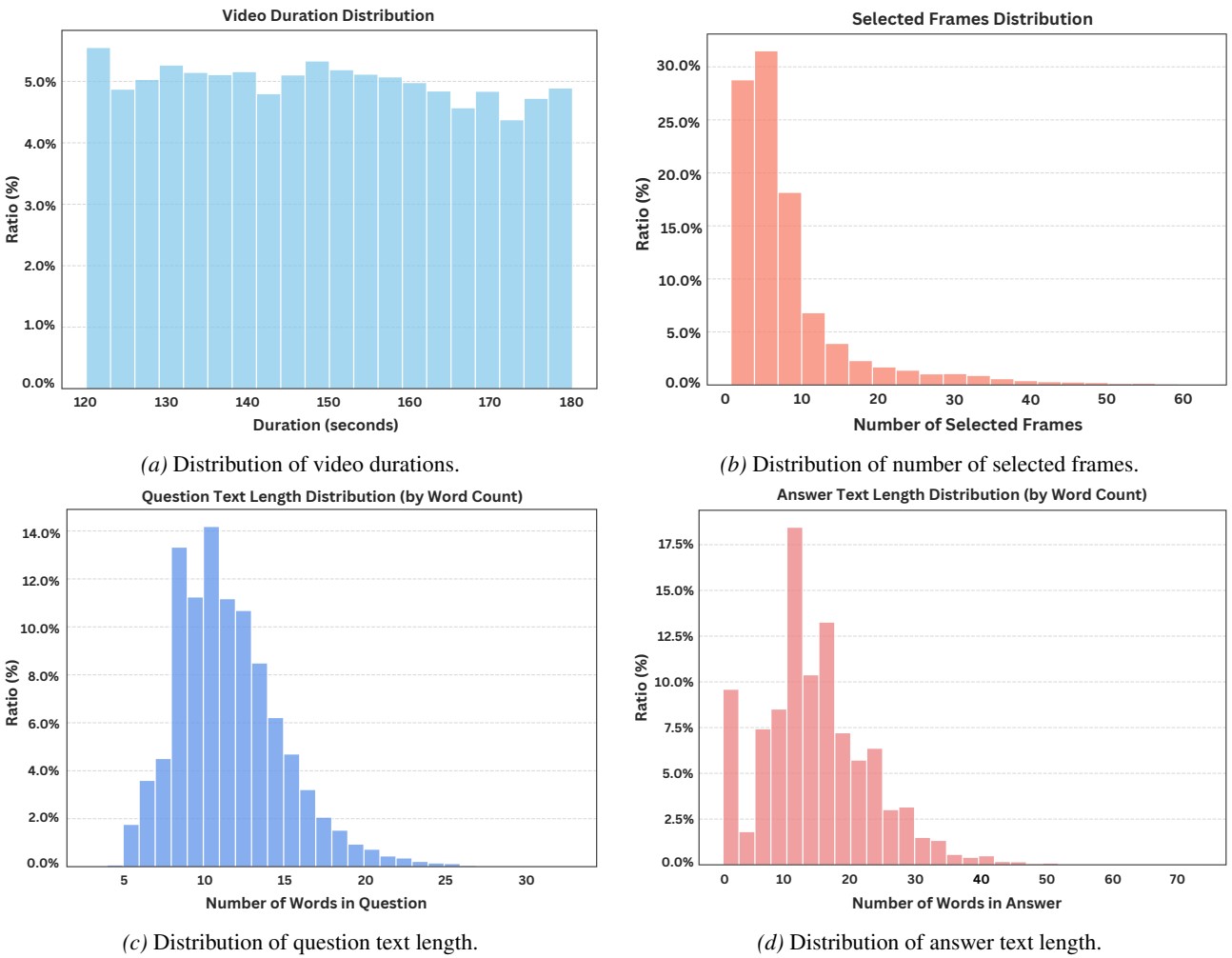

*(a)* Distribution of video durations.

*(b)* Distribution of number of selected frames.

*(c)* Distribution of question text length.

*(d)* Distribution of answer text length.

*Figure 5.* **Additional dataset statistics.** (a) Distribution of video durations. Most videos are between two and three minutes long. (b) Distribution of the number of selected keyframes per video–question pair. (c) Distribution of question lengths. Most questions are short, typically ranging from 6 to 15 words. (d) Distribution of answer lengths. Answers show a slightly higher variance, indicating that some responses are longer or more descriptive.

context for reasoning while avoiding excessive redundancy. Figure 5b shows the distribution of the number of selected keyframes per video–question pair: the median is five frames, the mean is around seven, and over 80% of instances require no more than 10 frames, with a small fraction of more complex cases requiring substantially more frames. Together, these distributions illustrate that most questions can be answered with a compact set of frames, while a minority demand broader temporal coverage.

Figure 5c shows that most questions are short, typically between six and fifteen words, peaking around ten. This reflects our design goal of keeping each question focused on a single aspect of the video. Figure 5d shows answer lengths, which are generally similar to the questions but slightly more variable: most are short, yet some extend into longer, descriptive phrases. Together, these figures indicate that questions and answers are mostly compact, while answers allow some variation, supporting models in producing both concise labels and richer, sentence-level responses.

To assess whether the two-stage data construction process introduces substantial shifts in dataset composition, we analyze the distribution of question types before and after filtering. Figure 6 provides a sanity check on this process by comparing the question-type distribution before and after the two-stage construction pipeline (Stage I: agent-based keyframe mining; Stage II: multi-VLM verification). Across categories, the absolute change in proportion is modest, with no category exhibiting a drastic increase or collapse.

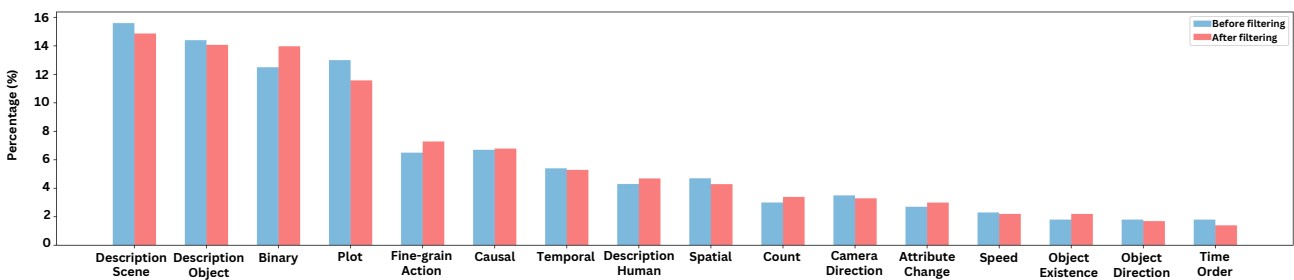

*Figure 6.* Question type distribution before and after the two-stage data construction process used to build FrameOracle-41K (Stage I: agent-based keyframe mining; Stage II: multi-VLM verification). All 16 question types remain represented, and the overall distribution structure is preserved.

## E. Human Verification of FrameOracle-41K Annotations

| Annotator Agreement Category | Count | Percentage |
|---|---|---|
| Both correct | 3,732 | 93.3% |
| One correct / one wrong | 240 | 6.0% |
| Both wrong | 28 | 0.7% |

*Table 9.* Distribution of pairwise annotation outcomes on the human verification set (4,000 samples). A sample is classified as "Both correct" if both annotators answered the question correctly using the provided keyframes, "One correct / one wrong" if exactly one annotator answered correctly, and "Both wrong" if neither annotator answered correctly.

| Category | Criterion | Count | Percentage |
|---|---|---|---|
| Excessive | $\geq$4 frames beyond minimal sufficiency | 118 | 3.1% |
| Just-right | within $\pm$3 frames of minimal sufficiency | 3,608 | 96.7% |
| Insufficient | Missing critical evidence | 6 | 0.2% |

*Table 10.* **Human verification on frame sufficiency.** We classify the annotated keyframe sets into three categories based on the gap between the annotated count and the human-perceived minimal sufficiency.

To ensure the reliability of FrameOracle-41K's automatically generated annotations, we conduct a human verification study on 4,000 randomly sampled instances ($\approx 10\%$ of the dataset). Ten independent annotators participate, with each sample reviewed by two distinct annotators. A sample is considered correct only if both annotators can answer the question using only the provided keyframes, without access to the full video or ground-truth answer. Inter-annotator agreement is high, at 94%, indicating strong consistency. As shown in Table 9, the overall human-verified accuracy is 93.3% (3,732/4,000), confirming that the vast majority of mined keyframe sets provide sufficient evidence for human-level reasoning.

Beyond correctness, we assess whether the annotated frames follow the principle of "minimal sufficiency." For all verified samples, annotators label keyframe sets as *Excessive*, *Just-right*, or *Insufficient*. Table 10 provides detailed definitions and statistics. As shown, the vast majority of samples (96.7%) are *Just-right*, with only a small fraction classified as excessive or insufficient. This confirms that FrameOracle-41K provides high-quality supervision that is both semantically accurate and frame-efficient, validating the effectiveness of our automated multi-stage generation pipeline.

## F. Additional Ablation Studies and Analysis

In this section, we present ablation studies and comparative analyses to validate FrameOracle's architectural and methodological choices. We examine eight key aspects to highlight the robustness and efficiency of our approach: (1) the **impact of the supervision backbone**, demonstrating that the selector generalizes across different teacher models and effectively leverages stronger visual representations to enhance long-context reasoning; (2) the **source of performance gains**, confirming through equal-budget comparisons that improvements come from selecting semantically relevant frames rather than simply reducing visual input; (3) the **effect of segment-wise candidate sampling for extremely long videos**, showing that splitting a long video into temporal segments and applying FrameOracle independently to each segment can effectively expose the selector to more frames without retraining; (4) the **sensitivity to the frame-cost coefficient**, showing that the value used in our main

experiments lies in a stable operating region and provides a smooth accuracy–efficiency trade-off; (5) the **stability of freezing the Rank Head**, verifying that ranking consistency is preserved in Stage 3 and fully refined by subsequent fine-tuning in Stage 4; (6) the benefits of **frame-level over token-level budgeting**, demonstrating that maintaining whole-frame integrity outperforms rigid token limits; (7) the advantage of **explicit selection over memory compression**, with substantial gains over long-video compression baselines; and (8) the **effect of motion dynamics on adaptive frame budgets**, showing that FrameOracle selects more frames for high-motion videos and fewer frames for low-motion videos. Together, these results show that FrameOracle's effectiveness stems from intelligent, query-conditioned semantic frame selection.

### F.1. Impact of the Supervision Backbone

| Model | Frames | NExTQA | | | Perception | LVB | Video-MME | EgoSchema | MLVU |
|---|---|---|---|---|---|---|---|---|---|
| | | OE_val | OE_test | MC | | | | | |
| LLaVA-OneVision-7B | 16 | 14.6 | 16.7 | 78.2 | 56.4 | 55.0 | 56.1 | 60.8 | 60.9 |
| **+ FrameOracle (Qwen2.5-VL-3B)** | 13.9 | 16.5 | **19.0** | **78.5** | 56.9 | 56.5 | 58.1 | 63.4 | 63.7 |
| **+ FrameOracle (VideoLLaMA3-7B)** | 14.2 | **16.7** | 18.9 | **78.5** | **57.0** | **57.4** | **58.4** | **64.0** | **65.1** |

*Table 11.* Comparison of FrameOracle models trained with different VLM backbones (Stage 2/3).

In our default setting, Stages 2 and 3 use Qwen2.5-VL-3B to provide soft supervision (via VLM loss) for training the Rank Head and K Head. To assess whether FrameOracle relies on specific architectural biases of the teacher model, we conduct an ablation by replacing Qwen2.5-VL-3B with the more powerful VideoLLaMA3-7B during training. To ensure a controlled comparison, both selector variants are evaluated using the same downstream pipeline (LLaVA-OneVision-7B) on the same benchmarks.

As shown in Table 11, replacing the backbone has minimal effect on the predicted frame counts, with both selectors producing nearly identical values (13.9 versus 14.2). Crucially, both variants outperform the full-frame baseline, demonstrating that FrameOracle generalizes well across different visual backbones. Furthermore, the selector trained with VideoLLaMA3-7B achieves stronger performance across benchmarks. This improvement stems from VideoLLaMA3-7B's stronger visual representations, which provide richer information about events spanning longer periods; as a result, this selector is better at choosing frames that capture high-level or long-range context, leading to larger gains on long-video benchmarks (e.g., MLVU).

### F.2. Do FrameOracle's Gains Come from Fewer Frames or Better Frames?

| Model | Frames | NExTQA | | | Perception | LVB | Video-MME | EgoSchema | MLVU | Avg. |
|---|---|---|---|---|---|---|---|---|---|---|
| | | OE_val | OE_test | MC | | | | | | |
| LLaVA-OneVision-7B | 16 → 10 (Uniform) | 8.8 | 11.5 | 73.5 | 53.8 | 53.2 | 54.1 | 60.0 | 55.8 | 46.3 |
| **+ FrameOracle** | **16 → 10.4** | **16.1** | **17.8** | **77.6** | **56.5** | **55.5** | **56.0** | **62.4** | **60.2** | **50.3** |

*Table 12.* Evaluation on LLaVA-OneVision-7B under an equal frame budget.

We assess whether FrameOracle's improvement over the full-frame baseline comes from better frame selection rather than simply using fewer frames. To do this, we compare FrameOracle with a uniform sampling baseline under the same frame budget. Using LLaVA-OneVision-7B, FrameOracle reduces 16 input frames to an average of 10.4. We then uniformly sample 10 frames from the same 16-frame inputs and evaluate both on the same benchmarks. As Table 12 shows, uniform sampling achieves 46.3% average accuracy, while FrameOracle reaches 50.3%. This 4.0-point gain confirms that the improvement comes from selecting semantically relevant, query-conditioned frames, not merely from reducing the number of frames.

### F.3. A Segment-wise Extension of FrameOracle for Long Videos

In extremely long videos, uniformly sampling 64 candidate frames from the entire video may provide insufficient temporal coverage, as it can miss critical evidence before any frame selection is applied. Since FrameOracle is trained with a fixed-size input, we apply a simple inference-time strategy that allows the same trained Rank Head to be applied to different temporal segments of a long video, thereby covering more frames in total without retraining. Specifically, we compare the

| Model | Frames | MLVU |
|---|---|---|
| LLaVA-OneVision-7B | 8 | 58.4 |
| + FrameOracle | $64 \rightarrow 8$ | 62.9 |
| **+ FrameOracle (Segmented)** | $128 \rightarrow 8$ | **65.4** |
| Qwen2.5-VL-7B | 8 | 54.5 |
| + FrameOracle | $128 \rightarrow 8$ | 59.6 |
| **+ FrameOracle (Segmented)** | $256 \rightarrow 8$ | **62.1** |

*Table 13.* **Ablation comparing single-pass sampling and two-segment sampling on MLVU.** Segment-wise inference applies the same trained 64-frame FrameOracle to two temporal segments and aggregates the top-4 frames per segment.

default setting using the 64-frame FrameOracle variant with a segment-wise sampling setting designed to increase temporal coverage. In this setting, the video is partitioned into two non-overlapping temporal segments, and 64 candidate frames are uniformly sampled from each segment. The same trained Rank Head is applied independently within each segment to rank frames conditioned on the query, and the top-4 frames from each segment are selected. The selected frames are then aggregated to form the final 8-frame input to the downstream VLM.

As shown in Table 13, increasing candidate exposure at inference time via segment-wise sampling consistently improves performance on MLVU. For both LLaVA-OneVision-7B and Qwen2.5-VL-7B, applying FrameOracle on two temporal segments yields clear gains over the single-pass setting, despite using the same trained 64-frame Rank Head and keeping the final input size fixed. These results indicate that, in extremely long videos, the primary limitation lies in insufficient candidate coverage rather than the selector's ranking capability. Once relevant evidence is observed, FrameOracle can reliably prioritize informative frames, and exposing it to a larger portion of the video at inference time is sufficient to recover missed cues without retraining or increasing the downstream frame budget.

### F.4. Sensitivity to the Frame-Cost Coefficient $\lambda$

| $\lambda$ | Avg. Predicted $K$ | Avg. Acc. $\uparrow$ | Latency (s) $\downarrow$ |
|---|---|---|---|
| 0.003 | 15.9 | 55.0 | 0.769 |
| 0.010 | 10.9 | 54.7 | 0.401 |
| **0.0105** | **10.4** | **54.7** | **0.363** |
| 0.012 | 8.3 | 54.3 | 0.298 |
| 0.015 | 4.5 | 52.6 | 0.201 |
| 0.030 | 1.2 | 46.1 | 0.107 |

*Table 14.* Sensitivity analysis of the frame-cost coefficient $\lambda$ in Eq. (2). We evaluate the 16-frame FrameOracle variant using LLaVA-Video-7B as the downstream inference backbone.

We evaluate the sensitivity of FrameOracle to the frame-cost coefficient $\lambda$ in Eq. (2), which controls the trade-off between downstream task loss and the number of selected frames when constructing the Stage 3 target frame count. Specifically, we evaluate $\lambda \in \{0.003, 0.010, 0.0105, 0.012, 0.015, 0.030\}$ using the 16-frame FrameOracle variant with LLaVA-Video-7B as the downstream inference backbone. The value $\lambda = 0.0105$ is used in our main experiments. As shown in Table 14, $\lambda$ provides a clear and interpretable control over the predicted frame budget. Smaller values place less penalty on frame usage and therefore retain more frames, while larger values encourage more aggressive compression. Around the value used in our main experiments, FrameOracle remains stable: nearby settings such as $\lambda = 0.010$ and $\lambda = 0.012$ yield similar accuracy while smoothly trading off latency and the number of selected frames. This indicates that the main setting does not rely on a brittle hyperparameter choice.

We also observe that extreme values push the selector toward boundary regimes. When $\lambda$ is too small, the model retains nearly all 16 frames, reducing the efficiency benefit. When $\lambda$ is too large, the model selects too few frames, which substantially hurts accuracy. These results confirm that Eq. (2) behaves as intended: $\lambda$ controls the efficiency-accuracy trade-off, and the value used in our main experiments lies in a stable operating region.

## F.5. Effect of Freezing the Rank Head in Stage 3

We evaluate how freezing the Rank Head during Stage 3, while updating the Transformer encoder layers, affects its alignment with the evolving frame representations. This alignment is crucial for accurately ranking frames, especially in long videos where subtle differences in importance matter. To assess this, we perform a dedicated ablation measuring both (1) ranking consistency and (2) downstream task performance.

| Model | Kendall-$\tau$ (vs. Ground Truth) |
|---|---|
| Stage 2 | 0.5367 |
| Stage 3 | 0.5221 |
| Stage 4 | **0.5833** |

*Table 15.* Kendall-$\tau$ consistency across training stages.

**Frame-Importance Consistency.** We randomly sample 500 training videos and compute a "ground-truth" importance distribution using leave-one-out (LOO) VLM loss. We then evaluate how well the Rank Head from Stages 2, 3, and 4 aligns with this distribution using Kendall-$\tau$ correlation. A large drop from Stage 2 to Stage 3 would indicate that freezing the Rank Head reduces its ability to track the encoder's evolving representations. As Table 15 shows, $\tau$ decreases only slightly after Stage 3. This is largely due to the very low learning rate ($1 \times 10^{-7}$) used for encoder updates, which limits feature drift. Stage 4 fully restores and even improves the alignment, demonstrating that any temporary misalignment is easily corrected through supervised fine-tuning.

| Model | Frames | NExTQA | | | Perception | LVB | Video-MME | EgoSchema | MLVU | Avg. |
|---|---|---|---|---|---|---|---|---|---|---|
| | | OE_val | OE_test | MC | | | | | | |
| Stage 2 | 32→16 | 24.8 | 29.5 | 73.0 | 64.7 | 51.9 | 55.7 | 52.2 | 54.8 | 50.8 |
| Stage 3 | 32→16 | 24.8 | 29.5 | 72.7 | 64.9 | 52.0 | 56.0 | 51.8 | 54.4 | 50.8 |
| Stage 4 | 32→16 | **25.3** | **30.0** | **74.0** | **65.8** | **52.9** | **56.9** | **52.8** | **55.5** | **51.7** |

*Table 16.* Downstream performance across training stages under a fixed 16-frame selection.

**Downstream Benchmark Evaluation.** We assess the practical impact of freezing the Rank Head in Stage 3 by evaluating Stages 2, 3, and 4 on video benchmarks using the Qwen2.5-VL-3B backbone with 32-frame inputs. To ensure a controlled comparison, we fix the number of selected frames to 16 across all stages, isolating the effect of ranking drift. As shown in Table 16, Stage 3 performs on par with Stage 2, indicating that the temporary freeze causes only negligible drift. Stage 4 consistently boosts performance across all benchmarks, confirming that supervised fine-tuning effectively realigns and strengthens the ranking behavior. These results demonstrate that temporarily freezing the Rank Head does not meaningfully harm the selector, even in long-video scenarios where ranking stability is critical.

| Model | Information | Perception | VideoMME | EgoSchema |
|---|---|---|---|---|
| VideoLLaMA2+B-VLLM | 1fps → 512 tokens | 48.0 | 44.4 | 44.3 |
| VideoLLaMA2+FrameOracle | 64 → 12.9 frames | **53.4** | **54.1** | **51.8** |
| B-VLLM | 1fps → 512 tokens | 52.1 | 53.5 | 51.9 |
| LLaVA-Video+FrameOracle | 64 → 12.9 frames | **65.1** | **61.6** | **55.2** |

*Table 17.* Comparison of information budgeting strategies: Frame-level (FrameOracle) vs. Token-level (B-VLLM (Lu et al., 2025)).

## F.6. Frame-level vs. Token-level Information Budgeting

An important question in adaptive video understanding is how to define the unit of computational budget: by visual tokens or by frames. To investigate this, we compare two representative strategies:

- **Token-level Budgeting:** Operates at a fine-grained level, selecting specific spatial-temporal tokens to fit a **fixed** total budget. This may discard parts of a frame to save computation.

- **Frame-level Budgeting (Ours):** Operates at a coarser granularity, treating each frame as an atomic unit. It **dynamically predicts** a budget of $K$ full frames, preserving the complete spatial context of each selected frame.

To compare these strategies, we select B-VLLM (Lu et al., 2025) as a representative token-level budgeting method. B-VLLM samples videos at 1 fps and enforces a fixed budget of 512 visual tokens, selecting the most relevant spatial tokens across the sequence. We compare this against FrameOracle, which predicts a dynamic number of frames. For a fair comparison, we evaluate both methods under a unified backbone (VideoLLaMA2 (Cheng et al., 2024)) to isolate the effect of the selection mechanism. Additionally, we compare B-VLLM's best-reported performance with our LLaVA-Video integration, both using Qwen2 (Yang et al., 2024) as the backbone LLM. As shown in Table 17, FrameOracle consistently outperforms B-VLLM across all benchmarks. These results indicate that preserving the spatial integrity of frames is crucial for reasoning and that dynamically selecting the number of frames is a more effective approach for controlling visual information than enforcing a fixed token budget.

### F.7. Comparison with Long Video Compression Methods

| Model | Frame | MovieChat-1K | | LVB | VideoMME | EgoSchema |
|---|---|---|---|---|---|---|
| | | Acc. | Score | | | |
| *(1) Reference Methods* | | | | | | |
| MovieChat (Song et al., 2024) | 2048 | 62.3 | 3.81 | - | - | - |
| MovieChat+ (Song et al., 2025) | 2048 | 71.2 | 3.51 | - | - | - |
| ReWind (Diko et al., 2025) | 548 | 80.6 | 4.46 | - | - | - |
| *(2) Controlled Comparison (Backbone: LLaVA-OneVision)* | | | | | | |
| + MovieChat | $512 \rightarrow 64$ | 67.1 | 3.44 | 44.2 | 45.6 | 57.8 |
| **+ FrameOracle (Ours)** | $64 \rightarrow 15.6$ | **69.6** | **3.82** | **56.5** | **58.1** | **63.4** |

*Table 18.* **Comparison with long video compression methods.** Note that reference methods do not report results on newer benchmarks (indicated by -). Best performance of reference methods is indicated by underline.

We further compare FrameOracle with memory-based compression methods designed for long video understanding. Unlike frame-selection approaches, methods such as MovieChat (Song et al., 2024), MovieChat+ (Song et al., 2025), and ReWind (Diko et al., 2025) process video streams sequentially, maintaining a learnable or fixed-size memory buffer that compresses historical frame features to control context length. For a fair comparison, we reproduce MovieChat using the same backbone as FrameOracle (LLaVA-OneVision). Following the official implementation, the reproduced MovieChat processes up to 512 input frames, compressing them into a maximum of 64 frames for the downstream VLM. We use the 64-frame version of FrameOracle for this comparison.

Table 18 summarizes the results on the MovieChat-1K benchmark and other standard long-video benchmarks. For reference, we also report the originally published numbers for MovieChat, MovieChat+, and ReWind. On MovieChat-1K, FrameOracle achieves 69.6% accuracy, showing that adaptive keyframe selection can match or surpass dense-frame baselines while using far fewer frames. On long-video benchmarks such as LongVideoBench, VideoMME, and EgoSchema, FrameOracle consistently outperforms compression-based methods, improving accuracy by up to +12.3% on LongVideoBench. These results indicate that adaptively selecting a small set of semantically rich keyframes provides stronger supervision and better generalization than compressing long frame sequences into fixed tokens or memory buffers.

### F.8. Effect of Motion Dynamics on Adaptive Frame Budgets

| Model | Motion Group | Avg. Selected Frames | LongVideoBench |
|---|---|---|---|
| FrameOracle | Low Motion | 24.6 | 69.9 |
| FrameOracle | High Motion | 31.0 | 68.1 |

*Table 19.* Effect of motion dynamics on FrameOracle's adaptive frame budgets.

We further analyze how FrameOracle adjusts its frame budget under different video dynamics. For each video in LongVideoBench, we estimate a motion score by computing the average cosine distance between adjacent DINOv2 frame embeddings. We then split the benchmark into low-motion and high-motion subsets and evaluate FrameOracle using Qwen3-VL-8B as the downstream backbone. As shown in Table 19, FrameOracle selects fewer frames for low-motion videos and more frames for high-motion videos. Specifically, the average selected frame count increases from 24.6 on low-motion videos to 31.0 on high-motion videos. This indicates that the K Head adapts the frame budget according to the temporal dynamics of the input: videos with more visual changes typically require broader temporal evidence, while

more static videos can be represented with fewer frames. At the same time, FrameOracle maintains strong performance on both subsets, achieving 69.9 on low-motion videos and 68.1 on high-motion videos. These results provide additional evidence that FrameOracle's adaptive selection behavior is not simply a fixed compression rule but reflects the amount of visual evidence needed for different video dynamics.

## G. Additional Benchmarks Details

| Benchmark | Response formatting prompts |
|---|---|
| MLVU | – |
| Video-MME | Answer with the option's letter from the given choices directly. |
| EgoSchema | Answer with the option's letter from the given choices directly. |
| NExTQA | – |
| Perception | Answer with the option's letter from the given choices directly. |
| LongVideoBench | Answer with the option's letter from the given choices directly. |

*Table 20.* Prompts specifying the response format used for each evaluation benchmark.

Table 20 summarizes the evaluation prompts for each benchmark used in our experiments, most of which are adapted from LMMs-Eval.

## H. FrameOracle-41K Examples

Figure 7 shows three examples from the FrameOracle-41K dataset. Each example demonstrates the number of selected keyframes, the associated question, the ground-truth answer, and the question type. We also provide indices for the selected keyframes within the 64 uniformly sampled frames used during preprocessing, indicating their relative positions along the video timeline. These examples highlight the diversity of reasoning types in FrameOracle-41K, such as causal reasoning and fine-grained action understanding, and illustrate that the annotations focus on semantically informative moments rather than evenly spaced frames.

## I. Qualitative Examples

As shown in Figure 8, FrameOracle can achieve correct answers using far fewer frames than uniform sampling. In the illustrated examples, our selector retains only 2–4 frames out of the original 16 inputs, yet these frames provide sufficient evidence to answer the questions accurately. This highlights that many uniformly sampled frames are redundant and that FrameOracle effectively filters them without sacrificing accuracy.

Figure 9 presents cases related to RQ1 (Section 5.2). Providing all 16 uniformly sampled frames can introduce irrelevant or distracting content, leading the VLM to produce incorrect answers. In contrast, FrameOracle selects a smaller, query-focused subset, allowing the model to concentrate on relevant evidence and answer correctly. These examples illustrate that more frames do not necessarily improve performance, and adaptive selection of fewer, informative frames enhances understanding.

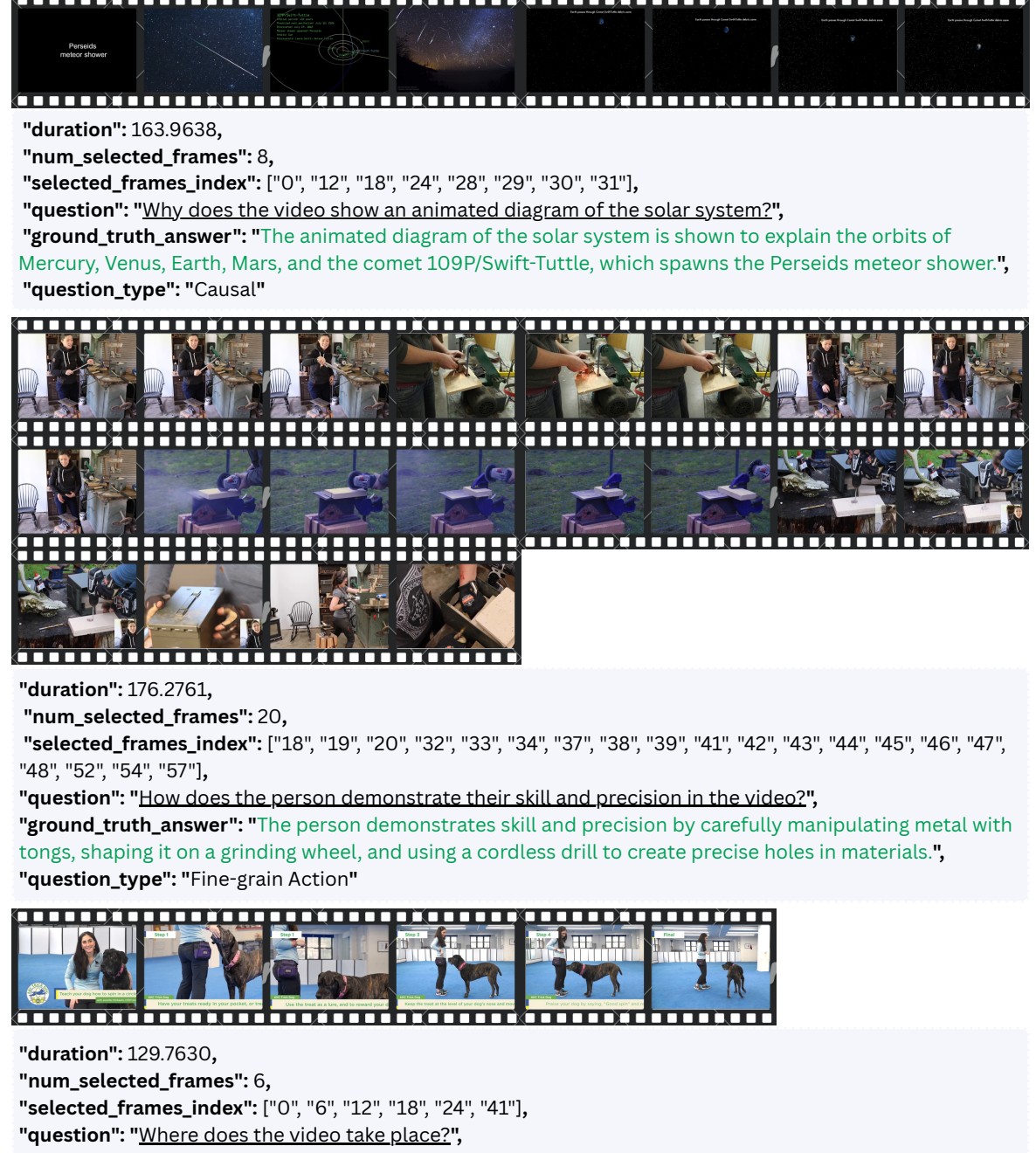

"duration": 163.9638,
"num_selected_frames": 8,
"selected_frames_index": ["0", "12", "18", "24", "28", "29", "30", "31"],
"question": "Why does the video show an animated diagram of the solar system?",
"ground_truth_answer": "The animated diagram of the solar system is shown to explain the orbits of Mercury, Venus, Earth, Mars, and the comet 109P/Swift-Tuttle, which spawns the Perseids meteor shower.",
"question_type": "Causal"

"duration": 176.2761,
"num_selected_frames": 20,
"selected_frames_index": ["18", "19", "20", "32", "33", "34", "37", "38", "39", "41", "42", "43", "44", "45", "46", "47", "48", "52", "54", "57"],
"question": "How does the person demonstrate their skill and precision in the video?",
"ground_truth_answer": "The person demonstrates skill and precision by carefully manipulating metal with tongs, shaping it on a grinding wheel, and using a cordless drill to create precise holes in materials.",
"question_type": "Fine-grain Action"

"duration": 129.7630,
"num_selected_frames": 6,
"selected_frames_index": ["0", "6", "12", "18", "24", "41"],
"question": "Where does the video take place?",
"ground_truth_answer": "The video takes place in an indoor training facility.",
"question_type": "Description Scene"

*Figure 7.* **Examples from the FrameOracle-41K dataset.** Each example shows the number of selected keyframes, question, ground-truth answer, question type, and the indices of the selected keyframes.

*Question*: What did the man in the front do when the man at the back after the man at the back picked up the spoon?

A. take the bowl

B. places pan back on stove

**C. dip food into sauce**

D. does hand gesture toward the tiger

E. help to season other chickens

**Uniform Sampling (Qwen2.5-VL: C)**

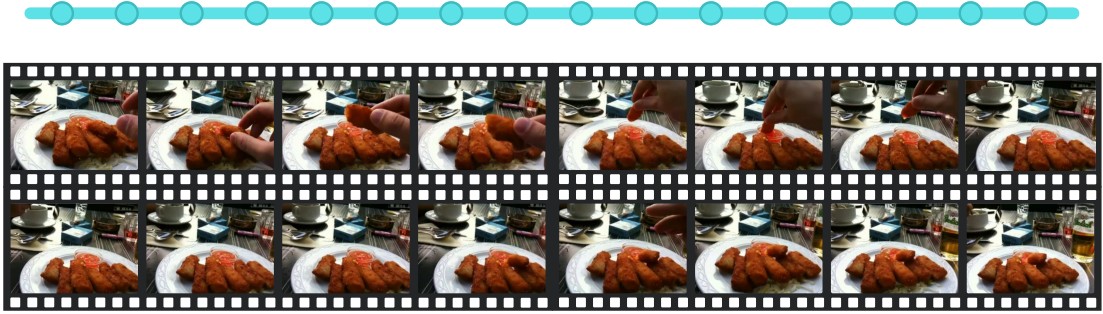

**FrameOracle (Qwen2.5-VL: C)**

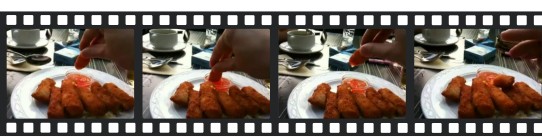

*Question*: Why is the man wearing slippers sitting at the top of the rock at the start of the video?

A. take photo from angle

B. sunbathing

C. preparing for a performance

D. to pose for the camera

**E. waiting to jump in water**

**Uniform Sampling (Qwen2.5-VL: E)**

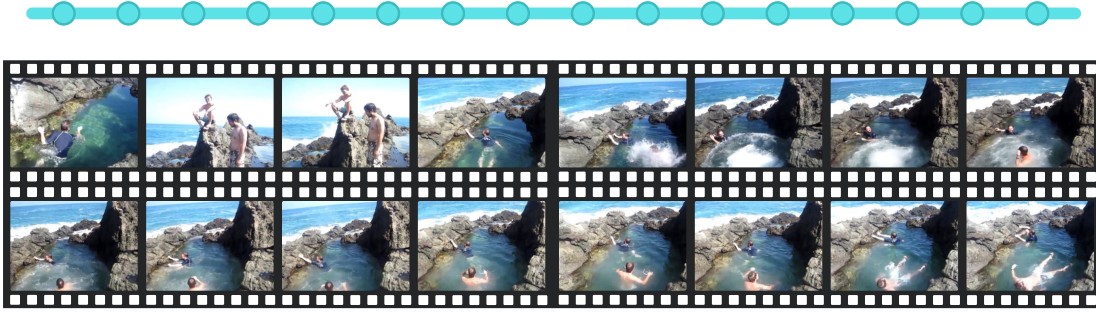

**FrameOracle (Qwen2.5-VL: E)**

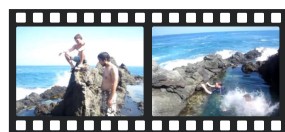

*Figure 8.* **Qualitative examples.** FrameOracle answers correctly while using only a few frames (2 to 4 out of 16), compared to uniform sampling, which relies on the full input.

_Question_: What seems to be the main purpose of the video? What actions did c perform to achieve this purpose?

A: The main objective of this instructional video is to effectively demonstrate how to easily tie your hair back.
B: The main purpose of the video is to show how to open a jar.
C: The primary objective of the video presentation is to demonstrate the most effective methods for properly cleaning your windows.
**D: The main purpose of the video is to show how to use a resistance band to exercise your arms and upper body.**
E: The primary objective of this video presentation is to effectively demonstrate the proper way to engage in a fun tug-of-war match with your canine companion.

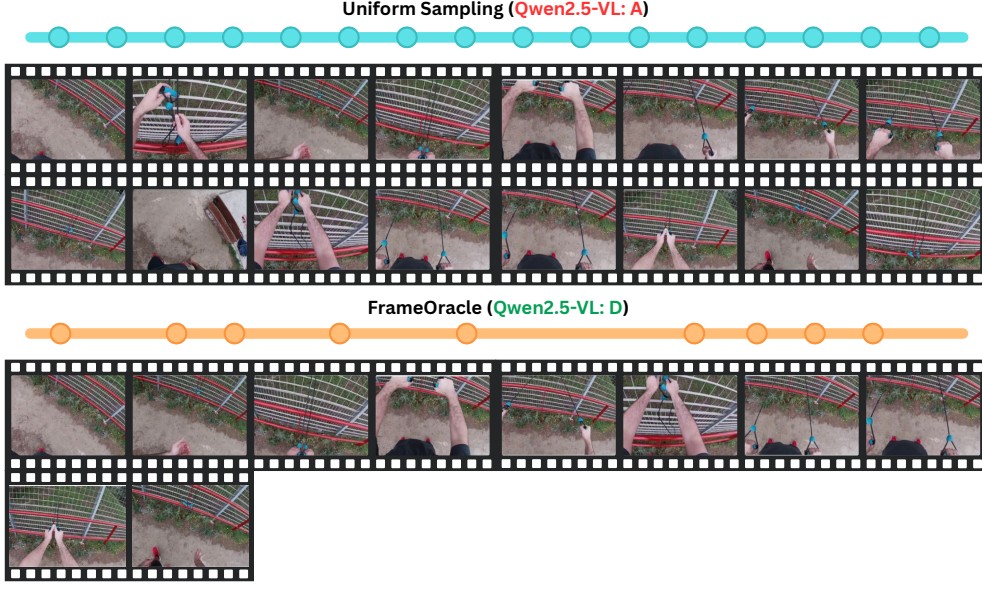

_Question_: From the sequence of actions, identify a turning point or moment where c's focus shifts to a different task. explain why you believe this is the most significant part of the video.

**A: The turning point is when c unfastens the hub axle.**
B: The crucial turning point occurs when character c picks up the screwdriver from the table.
C: The pivotal turning point occurs when character c decides to put on the gloves.
D: The turning point is when c removes the tire.
E: The critical turning point occurs when character c successfully patches the hole, fixing it.

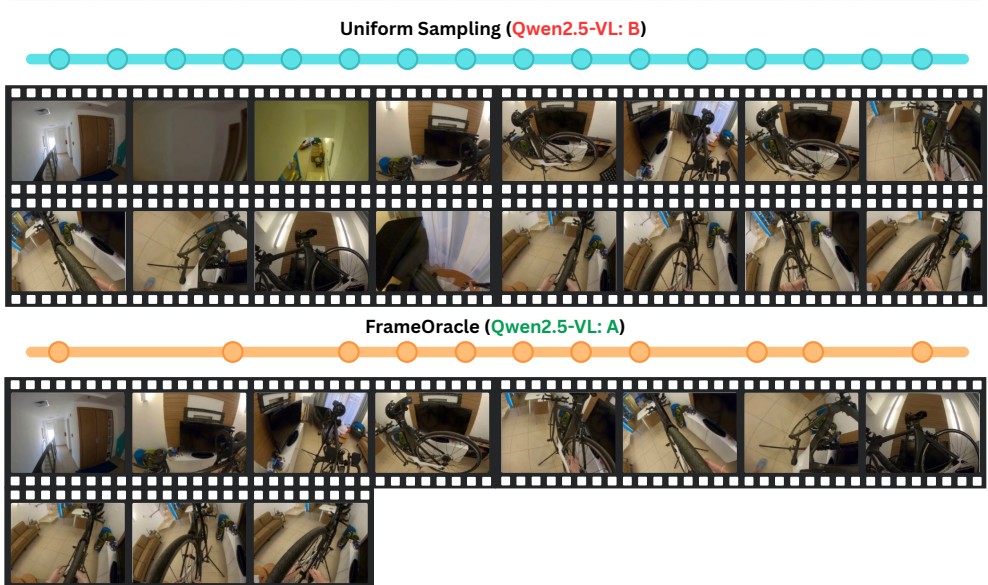

_Figure 9._ **Qualitative examples for RQ1.** Using all 16 uniformly sampled frames can produce incorrect answers, whereas FrameOracle answers correctly by selecting only the relevant subset.

