# OpenReview forum: "FrameOracle: Learning What to See and How Much to See in Videos"
_ICML.cc/2026/Conference — ICML 2026 regular_

### Official Review · Reviewer_QkT5 · 2026-03-02

**Soundness:** 3
**Presentation:** 3
**Significance:** 3
**Originality:** 3
**Overall Recommendation:** 4
**Confidence:** 4

**Summary:**

Authors propose FrameOracle, a lightweight, plug-and-play selector for video-VLMs that predicts both which frames are relevant to a given query and how many frames are actually needed, tackling the inefficiency of uniform or fixed-budget sampling. It’s trained via a four-stage curriculum that begins with weak proxy signals and culminates in supervised fine-tuning on a new dataset, FrameOracle-41K, which supplies keyframe annotations specifying the minimal sufficient frames per question. Across five VLMs and six benchmarks, FrameOracle cuts 16-frame inputs to ~10.4 with no accuracy loss and trims 64 candidates to ~13.9 while improving accuracy by ~1.4%.

**Compliance With Llm Reviewing Policy:**

Affirmed.

**Final Justification:**

The rebuttal addressed most of the concerns.

**Key Questions For Authors:**

This is not a weaknesses, just a curious question.

Question 1: The impact of different frame rates and resolutions of videos on this method.

Question 2: The impact of motion scenes and static scenes on candidate frames.

Question 3: Could you provide experimental results on blurred videos?

**Limitations:**

yes

**Strengths And Weaknesses:**

**Strengths**

1. Model-Agnostic and Plug-and-Play Design:
The proposed FrameOracle distinguishes itself through a model-agnostic architecture that operates independently of the underlying Video-Language Model (VLM). By eliminating the need for expensive co-training or model-specific fine-tuning, the method offers a highly practical, "plug-and-play" solution that can be seamlessly integrated into various existing VLM pipelines, ensuring strong transferability for real-world deployment.

2. Significant Data Contribution (FrameOracle-41K):
The introduction of the FrameOracle-41K dataset is a valuable contribution to the community. Unlike general video datasets, it provides fine-grained, keyframe-level supervision by identifying the minimal set of frames necessary to answer specific queries. This purpose-built resource fills a gap in VideoQA research, enabling more precise training and evaluation of adaptive frame selection strategies.

3. Superior Efficiency-Accuracy Trade-off:
The paper demonstrates that FrameOracle achieves a "Pareto improvement" in video understanding. It significantly reduces computational overhead by pruning redundant frames while simultaneously maintaining or even enhancing downstream task accuracy compared to full-frame baselines. This efficiency makes the approach particularly relevant for scaling VLMs to long-form video analysis.

4. Extensive and Robust Empirical Validation:
The effectiveness of FrameOracle is rigorously validated through a comprehensive suite of experiments. The authors evaluate the method across six diverse benchmarks and five different state-of-the-art VLMs. This consistent outperformance over existing keyframe selection baselines—in terms of both frame relevance and task-specific metrics—underscores the robustness and versatility of the proposed framework.

**Weaknesses**
While the proposed dataset is clearly a valuable resource, the core narrative of the paper focuses predominantly on its construction and benchmarking.
I believe the manuscript would receive better visibility and appreciation in a venue specifically dedicated to such contributions, such as the NeurIPS Datasets and Benchmarks Track.

---

> ### Author Rebuttal · Authors · 2026-03-30
>
> We thank the reviewer for the positive feedback and for highlighting the model-agnostic design, the contribution of FrameOracle-41K, and the strong efficiency–accuracy trade-offs demonstrated in our experiments.
>
> > While the proposed dataset is clearly a valuable resource, the core narrative of the paper focuses predominantly on its construction and benchmarking. I believe the manuscript would receive better visibility and appreciation in a venue specifically dedicated to such contributions, such as the NeurIPS Datasets and Benchmarks Track.
>
> We appreciate the comment regarding the dataset contribution. While FrameOracle-41K is an important component, the primary focus of this work is the design of a general, plug-and-play frame selection module (FrameOracle) that jointly models *what to see and how much to see*. The dataset enables this capability by providing supervision that is not available in existing benchmarks, but the method itself is designed to be broadly applicable across VLMs and datasets.
>
> > The impact of different frame rates and resolutions of videos on this method.
>
> To address this question, we conducted additional experiments on EgoSchema using Qwen3-VL-8B as the backbone.
>
> **(1) Spatial resolution.** We compared the original videos with a 480p version under the same evaluation setting. As shown below, reducing the resolution causes only a small drop for both the backbone (70.8 → 70.3) and FrameOracle (72.3 → 71.5). Importantly, FrameOracle still provides a clear improvement over the backbone at 480p (+1.2 points), indicating that its benefit does not rely heavily on high-resolution details.
>
>
> | Model | Resolution | EgoSchema |
> |---|---:|---:|
> | Qwen3-VL-8B   | Original | 70.8 |
> | + FrameOracle | Original | 72.3 |
> | Qwen3-VL-8B   | 480p     | 70.3 |
> | + FrameOracle | 480p     | 71.5 |
>
> **(2) frame rate.** We compared 30 fps and 15 fps versions under the same setting. As shown below, reducing the frame rate has minimal impact: the backbone changes from 70.8 to 70.6, and FrameOracle remains strong (72.3 → 72.2). The performance gain from FrameOracle is preserved in both cases (+1.5 at 30 fps and +1.6 at 15 fps).
>
>
> | Model | Frame Rate | EgoSchema |
> |---|---:|---:|
> | Qwen3-VL-8B   | 30fps | 70.8 |
> | + FrameOracle | 30fps | 72.3 |
> | Qwen3-VL-8B   | 15fps | 70.6 |
> | + FrameOracle | 15fps | 72.2 |
>
>
> > The impact of motion scenes and static scenes on candidate frames.
>
> To address this question, we conducted an additional experiment on LongVideoBench using Qwen3-VL-8B as the backbone. We estimate a motion score for each video by computing the average cosine distance between adjacent frame embeddings (from DINOv2), and split the dataset into **low-motion** and **high-motion** subsets.
>
> As shown below, FrameOracle selects fewer frames for low-motion videos than for high-motion videos (24.6 vs. 31.0), indicating that it **adapts its frame budget to the temporal dynamics of the video**. At the same time, performance remains comparable across the two subsets (69.9 vs. 68.1), demonstrating that the method handles both static and dynamic scenes effectively while adjusting how much visual evidence it retains.
>
> | Model | Motion Group | Frames | LongVideoBench |
> |---|---:|---:|---:|
> | FrameOracle | Low Motion  |  24.6  | 69.9 |
> | FrameOracle | High Motion |  31.0  | 68.1 |
>
> > Could you provide experimental results on blurred videos?
>
> To address this question, we conducted an additional experiment on EgoSchema using Qwen3-VL-8B as the backbone. We applied a Gaussian blur perturbation (kernel size=9 sigma=2.0) to the input videos while keeping the rest of the pipeline unchanged.
>
>
> As shown below, Gaussian blur causes a modest drop for both the backbone (70.8 → 69.6) and FrameOracle (72.3 → 71.3). Importantly, FrameOracle continues to provide a clear improvement under the blurred setting (+1.7 points), demonstrating that its benefit is preserved and that it remains **robust to visual degradation**.
>
>
> | Model | Blur | EgoSchema |
> |---|---:|---:|
> | Qwen3-VL-8B   | ✗ |  70.8    |
> | + FrameOracle | ✗ |  72.3    |
> | Qwen3-VL-8B   | ✓ |  69.6    |
> | + FrameOracle | ✓ |  71.3    |

---

> > ### Author Rebuttal · Reviewer_QkT5 · 2026-04-02
> >
> > The authors have directly responded to my concerns with quantitative results and clear discussions.

---

> > > ### Author Response · Authors · 2026-04-02
> > >
> > > Thank you very much for your follow-up. We are grateful that you found our rebuttal responsive and well supported by quantitative results and clear discussion. If you feel the main concerns have now been satisfactorily resolved, we would sincerely appreciate it if you could consider increasing your score accordingly.

---

### Official Review · Reviewer_UFfH · 2026-03-08

**Soundness:** 3
**Presentation:** 3
**Significance:** 2
**Originality:** 2
**Overall Recommendation:** 4
**Confidence:** 3

**Summary:**

The paper presents FrameOracle, a lightweight, plug-and-play module that predicts both which frames are most relevant to a given query and how many frames are needed. It is trained via a curriculum that progresses from weak proxy signals to stronger supervision with FrameOracle-41K, the first large-scale VideoQA dataset with validated keyframe annotations specifying minimal sufficient frames per question. Extensive experiments show that FrameOracle reduces 16-frame inputs to an average of 10.4 frames without accuracy loss and achieves state-of-the-art efficiency–accuracy trade-offs for scalable video understanding.

**Compliance With Llm Reviewing Policy:**

Affirmed.

**Final Justification:**

The rebuttal addresses several of my concerns, particularly by clarifying the distinctions from prior keyframe selection methods and providing an informative ablation on the hyperparameter λ. However, the overall novelty relative to existing approaches and some design choices (e.g., offline setting and reliance on pre-sampled frames) remain only partially addressed.

Given these points, I find the work solid but not substantially strengthened beyond the original assessment, and therefore I keep my score unchanged.

**Key Questions For Authors:**

How sensitive is the target frame number k to the hyperparameter λ in Eq. (2)? Could you provide an ablation study on different λ values?

**Limitations:**

1. An agent-based annotation pipeline is utilized for dataset processing. This may limit diversity in video content and question types, and could introduce systematic biases in the keyframe labels.

2. FrameOracle operates on pre-sampled candidate frames (16 or 64). It does not support online or incremental selection for streaming videos, which limits applicability to real-time scenarios.

**Strengths And Weaknesses:**

Strength
1. The paper is well-structured and easy to follow, with clear descriptions of the method and dataset.

2. The experiments are comprehensive, covering multiple VLMs and benchmarks, which strengthens the evaluation.

Weakness
1. The motivation for adaptive frame selection is solid, but the method shares similarities with existing keyframe approaches like KeyVideoLLM[1] or VCA[2], potentially limiting its novelty.

2. Figure 3 provides a good overview of the pipeline, but it could be improved by labeling data flows more explicitly and clarifying how the K Head integrates with the fusion encoder.

Overall, I recommend weak accept for this paper, as it offers useful contributions to video understanding efficiency, though some refinements in novelty and details are needed.

[1] Liang H, Li J, Bai T, et al. Keyvideollm: Towards large-scale video keyframe selection[J]. arXiv preprint arXiv:2407.03104, 2024.

[2] Yang Z, Chen D, Yu X, et al. Vca: Video curious agent for long video understanding[C] //Proceedings of the IEEE/CVF International Conference on Computer Vision. 2025: 20168-20179.

---

> ### Author Rebuttal · Authors · 2026-03-30
>
> We thank the reviewer for their thoughtful comments and for recognizing the clarity of the paper and the comprehensiveness of the experimental evaluation across multiple VLMs and benchmarks.
>
> > The motivation for adaptive frame selection is solid, but the method shares similarities with existing keyframe approaches like KeyVideoLLM or VCA ...
>
> FrameOracle differs from KeyVideoLLM and VCA in several key aspects.
>
> KeyVideoLLM is **non-adaptive and operates under a fixed frame budget** (i.e., always selecting top-k frames). In contrast, FrameOracle jointly models **which frames to keep and how many are needed**, enabling adaptive selection conditioned on the video–query pair.
>
> VCA, on the other hand, relies on **agent-based inference with repeated multimodal model calls (i.e., GPT-4o API)**, which can be computationally expensive in both latency and API cost. In contrast, FrameOracle learns an **80M-parameter lightweight plug-and-play module** that can be applied efficiently across different VLM backbones.
>
> Finally, FrameOracle is supported by FrameOracle-41K, which provides **question-conditioned keyframe annotations and minimal sufficient frame counts**, enabling direct supervision for both frame selection and frame-budget prediction.
>
> We will clarify these distinctions more explicitly in the final version.
>
> > Figure 3 provides a good overview of the pipeline, but it could be improved ...
>
> In the final version, we will revise Figure 3 to more explicitly label the data flow and intermediate representations, and make the branching structure more explicit by showing how the fused representation feeds both the ranking pathway and the K prediction pathway.
>
> > How sensitive is the target frame number k to the hyperparameter λ in Eq. (2)? Could you provide an ablation study on different λ values?
>
> We conducted an ablation study on the trade-off coefficient $\lambda$ in Eq. (2), which controls the balance between downstream task loss and frame cost when defining the Stage 3 target frame number. We evaluated $\lambda \in \lbrace 0.003, 0.010, 0.0105, 0.012, 0.0150, 0.030 \rbrace$ using the FrameOracle 16-frame variant with LLaVA-Video-7B as the inference backbone. The value $0.0105$ is used in our main experiments.
>
> As shown below, $\lambda$ provides a clear and consistent control over the predicted number of frames. Around the value used in our main experiments, the method shows stable behavior, with nearby settings (e.g., $0.0100–0.0120$) yielding similar performance.
>
> We also observed that when $\lambda$ becomes too small or too large, the predicted frame number collapses toward boundary cases, selecting nearly all 16 frames or nearly a single frame.
>
> | λ | Avg. predicted K | Avg. Acc. | Latency (s) |
> |---|---:|---:|---|
> | 0.0030 | 15.9 | 55.0 | 0.769 |
> | 0.0100 | 10.9 | 54.7 | 0.401 |
> | **0.0105** | **10.4** | **54.7** | **0.363** |
> | 0.0120 | 8.3 | 54.3 | 0.298 |
> | 0.0150 | 4.5 | 52.6 | 0.201 |
> | 0.0300 | 1.2 | 46.1 | 0.107 |
>
> > An agent-based annotation pipeline is utilized for dataset processing. This may limit diversity in video content and question types, and could introduce systematic biases in the keyframe labels.
>
> We take several steps to ensure diversity and reliability.
>
> FrameOracle-41K covers a wide range of scenarios, including **~41K video–question pairs** and **16 question types**, as well as varied video durations and diverse question and answer lengths, indicating substantial diversity in both video content and QA formulation (see Figure 2, Figure 5, and Appendix D).
>
> In addition, the final labels are not derived from a single agent pass. As described in Section 3.1, candidate keyframes are filtered through a **cross-model verification stage**, where three independent VLMs must correctly answer the question using only the selected frames. We further conduct **human verification** on 4,000 randomly sampled instances, achieving **94% inter-annotator agreement and 93.3% verified accuracy** (Appendix E).
>
> > FrameOracle operates on pre-sampled candidate frames and does not support streaming scenarios.
>
> The current design focuses on **offline frame selection** and does not directly support streaming or incremental settings. This choice is aligned with the problem setting in this paper, where FrameOracle operates on a pre-sampled candidate pool to perform query-conditioned frame selection under a constrained VLM budget.
>
> Streaming video understanding introduces additional challenges, such as causal decision-making without access to future frames, as well as stricter latency and memory constraints, and is typically studied under a different formulation [1]. We therefore view it as a complementary setting rather than a limitation of the current approach.
>
> Extending FrameOracle to support online or streaming selection is an interesting direction for future work.
>
> [1] Song et al. MovieChat: From Dense Token to Sparse Memory for Long Video Understanding. CVPR'24.

---

> > ### Author Rebuttal · Reviewer_UFfH · 2026-04-03
> >
> > Thank you for your response. I would keep my original score.

---

> > > ### Author Response · Authors · 2026-04-04
> > >
> > > Thank you very much for your thoughtful follow-up. We sincerely appreciate your time and are glad that our rebuttal has fully addressed your concerns.

---

### Official Review · Reviewer_GMSm · 2026-03-11

**Soundness:** 3
**Presentation:** 3
**Significance:** 3
**Originality:** 3
**Overall Recommendation:** 4
**Confidence:** 4

**Summary:**

This work introduces FrameOracle, a lightweight, plug-and-play module designed to enhance the efficiency and accuracy of Vision-Language Models (VLMs) through query-conditioned frame selection. Unlike static sampling methods, FrameOracle employs a backbone-agnostic selector that jointly predicts both the relevance of specific frames and the optimal sample density required for a given query. By adaptively filtering for high-signal visual data, FrameOracle significantly reduces the computational overhead of video analytics. Extensive evaluation across six benchmarks and five VLMs demonstrates that the module maintains—or even improves—accuracy while using a fraction of the original video frames.

**Compliance With Llm Reviewing Policy:**

Affirmed.

**Final Justification:**

The rebuttal adequately addresses my concerns. My recommendation is remain the same.

**Key Questions For Authors:**

(1) Related to point 2 in weakness, could the authors clarify the decision-making process behind the design of these stages? Specifically:
- Stage Determination: How were the specific datasets, data quantities, and the set of trainable modules for each stage (e.g., selection heads vs. backbone) determined?

- Heuristics vs. Empirical Search: Is the current configuration based on a high-level heuristic (e.g., starting with low-resolution or shorter temporal windows) or was it the result of an extensive empirical search across different joint-training schedules?

- Optimization Stability: Given that skipping stages leads to optimization instability, providing these insights would be invaluable for understanding the conditioning of the model’s loss landscape. A discussion on the 'transition criteria' between stages—and whether these were determined by loss convergence or a fixed epoch schedule—would greatly benefit the community in reproducing or extending this work.


(2) The data generation pipeline employs a sophisticated, multi-agent framework to exhaustively analyze frames and select keyframes. This multi-agent approach is inherently more reliable for novel or complex video-question pairs because it can cross-validate evidence across multiple "perspectives" (agents). While this serves as a high-quality "oracle" for training, there is a fundamental concern regarding the zero-shot robustness of the trained FrameOracle. It remains unclear how the FrameOracle may performs in zero-shot scenarios where the visual cues or question types deviate from the specific "reasoning paths" captured in the training data.

**Limitations:**

Yes

**Strengths And Weaknesses:**

Strengths
- FrameOracle-41K Dataset: The authors introduce the first large-scale VideoQA dataset with agent validated, question-grounded keyframe annotations. Unlike existing datasets that rely on weak proxy signals, this dataset provides explicit ground truth for the minimal sufficient frames required to answer a specific query. The dataset is statistically balanced, mitigates common biases, and leverages diverse video sources from LLaVA-Video-178K.

- Model-Agnostic Module: FrameOracle is designed as a highly flexible, plug-and-play pre-processing module. Its backbone-agnostic nature allows it to be integrated with various Vision-Language Models (VLMs) without the need for computationally expensive joint training or architectural modifications.

- Curriculum Training: The module’s effectiveness is underpinned by a structured four-stage training regime. This curriculum progressively transitions from coarse, cross-modal similarity signals to fine-grained, strong supervision using the FrameOracle-41K dataset, ensuring the selector learns both broad relevance and precise frame-level necessity.

Weaknesses
- Pre-sampling Overhead vs. Accuracy: A primary concern regarding FrameOracle is its dependency on the initial candidate frame coverage. For extremely long-form videos, maintaining high recall of "critical evidence" likely requires a significant increase in the pre-sampling rate. If the model requires a dense initial sweep to avoid missing sparse temporal cues, the cumulative computational cost may offset the efficiency gains of the subsequent selection process. To better contextualize the results in Table 3, the authors should include a comparison with a higher input budget (e.g., 32 or 64 frames). Presenting the trade-off between visual token count, latency, and memory overhead across these variants is essential. Without this, it is difficult to discern if FrameOracle provides a superior Pareto frontier for the "accuracy-vs-efficiency" trade-off compared to simply scaling the input capacity of existing state-of-the-art VLMs.

- Reliance on Multi-Stage Curriculum: The effectiveness of the proposed model appears heavily dependent on a specific, multi-stage curriculum. The experiment shows that attempting joint training of selection heads or skipping intermediate stages leads to optimization instability and a significant drop in performance. This sensitivity suggests a fragile optimization landscape, where the model may be prone to local minima or vanishing gradients if the curriculum is not precisely followed. It remains unclear if this training regime is robust across different video domains or if it requires extensive hyperparameter tuning for each new dataset.

---

> ### Author Rebuttal · Authors · 2026-03-30
>
> We thank the reviewer for their feedback and for recognizing the value of FrameOracle-41K, the plug-and-play design of FrameOracle, and the role of the multi-stage curriculum.
>
> > Pre-sampling Overhead vs. Accuracy...
>
> We extend Tab 3 with **raw 32-frame and 64-frame baselines**.
>
> As shown below, increasing the input budget yields **only modest accuracy gains but significantly higher cost**. For example, compared to FrameOracle (64→13.9), the raw 32-frame baseline improves accuracy by only +0.2, while requiring over **2× higher latency, FLOPs, and visual tokens**. Similarly, the 64-frame baseline improves accuracy by +0.6 but incurs nearly **5× higher compute and latency**.
>
> This shows that FrameOracle provides a more favorable **accuracy–efficiency trade-off** than increasing the input capacity alone.
>
> |Model|Frames|Total TFLOPs ↓|Latency (s) ↓|Visual Tokens ↓|Avg. Acc. ↑|
> |---|---:|---:|---:|---:|---:|
> |LLaVA-Video-7B|16|184.38|0.615|11,644.0|54.6|
> |LLaVA-Video-7B|32|405.64|1.140|23,290.0|56.2|
> |LLaVA-Video-7B|64|792.83|2.622|46,584.0|56.6|
> |+FrameOracle|64→13.9|167.67|0.556|10,133.1|56.0|
>
> > Reliance on Multi-Stage Curriculum...
>
> The staged design is motivated by the **dependency structure of the tasks**, rather than a fragile optimization recipe. **Accurate frame ranking must first be established** before learning frame-count prediction on top of it. When both heads are trained jointly from the beginning, they interfere with each other, and when intermediate stages are skipped, later stages lack the alignment and ranking signals they depend on (Appendix F1).
>
> Our experiments show that FrameOracle is **robust across domains without additional tuning**. In Tab 1, FrameOracle is applied across 6 benchmarks, ranging from short clips (~10sec) to long videos (2–3h), and covering a range of reasoning tasks, without curriculum redesign, hyperparameter tuning, or retraining. The reasoning tasks include fine-grained event recognition, local temporal reasoning, cross-event reasoning, global consistency, and long-range temporal grounding.
>
> > Stage Determination.
>
> The stage configuration is guided by the **type of supervision required at each stage** and follows a **coarse-to-fine learning paradigm**, rather than an extensive empirical search.
>
> Stage 1 uses large-scale video–question datasets (e.g., ShareGPT4o-Video, Video-ChatGPT) to learn general **text–visual alignment**, where broad coverage is more important than precise supervision. Stages 2–3 introduce datasets such as LLaVA-Hound, which are more aligned with **evidence-grounded video understanding**, to support learning frame selection under weak supervision. Stage 4 then uses FrameOracle-41K, where the model is trained with **direct supervision** on both keyframe identity and frame count.
>
> The choice of trainable and frozen modules is determined by the need to **isolate learning objectives and avoid interference**. For example, earlier stages focus on stabilizing representations and ranking, while later stages build on these representations to learn frame-count prediction (Details in Appendix A).
>
> > Heuristics vs. Empirical Search.
>
> The stage design is guided by a combination of **high-level heuristics and empirical validation**, rather than an extensive search over training schedules. We follow a **coarse-to-fine training paradigm**: first learning stable cross-modal representations, then refining frame ranking, and finally learning frame-count prediction.
>
> Specifically, Stage 1 uses text–visual alignment to initialize the encoder; Stages 2–3 learn ranking and frame-budget prediction under **proxy supervision**; and Stage 4 refines both using **direct supervision** from FrameOracle-41K. This design reflects the available supervision: large-scale datasets provide broad but weak signals, while FrameOracle-41K provides precise but smaller-scale annotations.
>
> > Optimization Stability. Transition between stages...
>
> The transition between stages is guided by **loss convergence**, rather than a fixed epoch schedule or an automatic threshold. In practice, we monitor the training loss and transition once it has **plateaued**, typically continuing for an additional **1–2 epochs** before moving to the next stage.
>
> > The data generation pipeline ...
>
> The multi-agent framework is used only to construct FrameOracle-41K, and **not to train FrameOracle to imitate agentic reasoning**. The model is trained solely on the final supervision signals (i.e., keyframe annotations and frame-count labels) without access to agent trajectories or intermediate decisions.
>
> Regarding robustness, we show **zero-shot transfer at inference time**. FrameOracle is applied in a plug-and-play manner across 5 VLM backbones and 6 benchmarks, without backbone-specific adaptation or tuning. These benchmarks span diverse video lengths and reasoning demands, indicating that FrameOracle learns a transferable frame-selection policy.

---

> > ### Author Rebuttal · Reviewer_GMSm · 2026-04-01
> >
> > The authors have directly responded to my concerns with quantitative results and clear discussions. I am leaning to accept this work.

---

> > > ### Author Response · Authors · 2026-04-01
> > >
> > > Thank you very much for your thoughtful follow-up. We are especially encouraged by your note that you are leaning to accept this work. If you feel the main concerns have been satisfactorily resolved, we would be very grateful if you could consider increasing your score accordingly.

---

### Official Review · Reviewer_p7U2 · 2026-03-11

**Soundness:** 3
**Presentation:** 3
**Significance:** 2
**Originality:** 2
**Overall Recommendation:** 4
**Confidence:** 4

**Summary:**

This work introduces FrameOracle, a lightweight and adaptable module that dynamically optimizes frame selection by identifying both the most task-relevant content and the ideal sequence length. To support this framework, the authors contribute FrameOracle-41K, a pioneering large-scale VideoQA dataset featuring minimal-set annotations that pinpoint the exact frames necessary to answer a query. Experimental results across multiple benchmarks and five state-of-the-art VLMs confirm that FrameOracle substantially reduces computational overhead while maintaining, or even improving, task accuracy.

**Compliance With Llm Reviewing Policy:**

Affirmed.

**Final Justification:**

The rebuttal addressed most of the concerns.

**Key Questions For Authors:**

- Fixed-FPS sampling may actually perform better than providing more frames, as MLLMs may struggle to effectively process large amounts of visual input. Could the authors clarify this point?
- Could the authors provide results when the proposed method is applied on existing keyframe selection baselines?

**Limitations:**

There is no limitation section.

**Strengths And Weaknesses:**

Strengths:
- The paper is clearly written and easy to follow.
- The paper introduces FrameOracle-41K, a novel large-scale VideoQA dataset annotated with the minimal subset of frames required to answer each query.
- By dynamically pruning redundant visual information, the proposed framework significantly improves computational efficiency compared to naive MLLM approaches.

Weaknesses:
- When compared with existing keyframe selection methods, the proposed approach does not show a significant performance improvement.
- The authors argue that existing keyframe selection methods relying on fixed FPS are a limitation; however, it is unclear why sampling a fixed number of frames across videos with varying durations would be more desirable.
- It is unclear how the baseline keyframe selection methods perform on the benchmarks reported in Table 1.
- The method consists of four stages, which introduces additional complexity to the overall framework.
- It would be useful to clarify whether FrameOracle-41K can provide additional benefits when used with other keyframe selection methods.

---

> ### Author Rebuttal · Authors · 2026-03-30
>
> We thank the reviewer for their thoughtful comments and for recognizing the clarity of the paper, the novelty of FrameOracle-41K, and the efficiency benefits of adaptive frame selection.
>
> > Comparison with existing methods...
>
> Comparison to existing methods in Tab 2 is done under a **fixed-budget setting (top-8 frames)**. To ensure a fair comparison, we **disable FrameOracle’s K Head** and evaluate only the Rank Head. This removes the adaptive frame-count prediction, which is a key component of our method.
>
> Even under this setting, FrameOracle achieves **competitive or better performance** than existing selectors across benchmarks.
>
> Enabling the full FrameOracle (Rank Head + K Head) leads to **consistent additional improvements**. An example on LLaVA-Video-7B is shown below:
>
>
> | Method | NExTQA | LongVideoBench| Video-MME | EgoSchema | MLVU |
> |---|---:|---:|---:|---:|---:|
> | + FrameOracle (64→8, Rank Head only)   | 76.5 | 56.9 | 58.9 | 53.0 | 63.4 |
> | + FrameOracle (64→14.1, full)| **81.6**|**57.8**|**61.6**|**55.2**|**64.3**|
>
>
> > Fixed-FPS vs fixed-count sampling.
>
> We do **not** claim that fixed-count sampling is preferable to fixed-FPS sampling. Rather, our point (L 13–19) is that **fixed-FPS and fixed-count methods are non-adaptive**, as neither adjusts to the specific video–query pair.
>
> FrameOracle instead jointly predicts **which frames to select** and **how many to retain**. This enables **adaptive** allocation of frames, leading to improved efficiency–accuracy trade-offs. Empirically, starting from 64 candidates, FrameOracle selects **13.9 frames on average** while improving accuracy over the fixed 16-frame baseline (Tab 1), showing that a fixed budget is suboptimal.
>
> The limitation of fixed-FPS sampling is that it applies a uniform temporal stride regardless of content density or query relevance. Fixed-count selection suffers from the same lack of adaptivity.
>
>
>
> > Baseline keyframe selection methods perform on the benchmarks reported in Table 1.
>
>
> Benchmarks are shared between Tab 1 and Tab 2. The only exception is Perception, which is not included in Tab 2 because prior methods do not report results on it.
>
> > The method consists of four stages, which introduces additional complexity.
>
> The multi-stage training is necessary to address the **lack of frame-level supervision** in existing datasets. FrameOracle-41K is the first dataset providing frame-level supervision, but with only 41K instances, it is much smaller than typical video-language corpora, making it difficult to train a selector from scratch without instability.
>
> We thus adopt a staged curriculum: first learning from **proxy supervision** (e.g., text–visual alignment and weak multimodal signals), and then refining with stronger supervision. Jointly optimizing these objectives from scratch leads to unstable convergence, as shown in our ablation.
>
> Once trained, FrameOracle is a lightweight **80M-parameter** module that can be directly applied as a **plug-and-play preprocessor** without additional cost or fine-tuning.
>
> > Use of FrameOracle-41K for other methods.
>
> Yes. FrameOracle-41K provides **frame-level keyframe annotations**, which can be used to supervise other methods. For example, a fixed-budget method can use the annotated keyframes as ground truth in an SFT-style setup, learning to rank relevant frames higher during training while retaining its original top-k selection at inference.
>
> Any method that predicts **frame importance** can benefit from this supervision, while methods that also predict **how many frames to retain** can further leverage the dataset’s frame-count annotations.
>
> > Fixed-FPS sampling may actually perform better than providing more frames ...
>
> FrameOracle is **not designed to increase the number of frames**, but to select a **smaller, query-relevant subset** (e.g., 64 → 13.9 frames).
>
> This also highlights a limitation of fixed-FPS sampling: it is query-agnostic and can still generate many unnecessary frames for long videos. For example, on LongVideoBench (~12 minutes), 1 FPS sampling produces ~720 frames. In practice, such methods still require additional truncation or selection. In contrast, FrameOracle directly selects a compact and relevant subset, achieving significantly fewer frames than pure fixed-FPS sampling.
>
> > Results when FrameOracle is applied on existing baselines?
>
> We evaluated FrameOracle with BOLT using Qwen2.5-VL-7B as the backbone.
>
> We first apply the full FrameOracle to obtain an adaptive candidate frame set for each video–question pair, and then apply BOLT to select the final 8-frame input. BOLT, like other baselines, is **non-adaptive** and operates under a fixed 8-frame budget, making this combination directly compatible.
>
> As shown below, **FrameOracle + BOLT consistently improves over BOLT** across all benchmarks.
>
> | Method | Video-MME | LongVideoBench | MLVU |
> |---|---:|---:|---:|
> | Qwen2.5-VL-7B|54.1|53.6|54.5|
> | BOLT|57.0|54.9|60.2|
> | FrameOracle + BOLT |**57.5**|**55.4**|**60.5**|

---

> > ### Author Rebuttal · Reviewer_p7U2 · 2026-04-04
> >
> > Thanks for the rebuttal. The concerns are well resolved.

---

> > > ### Author Response · Authors · 2026-04-04
> > >
> > > Thank you very much for your thoughtful follow-up. We sincerely appreciate your time and are glad that our rebuttal has fully addressed your concerns.

---

### Decision · Program_Chairs · 2026-04-30

**Decision:**

Accept (regular)

**Comment:**

In this paper, the authors presented a plug-and-play module for vision-language models (VLMs) and contributed a large-scale dataset. The paper was reviewed by four expert reviewers, followed by a rebuttal and discussion between the reviewers and authors. The paper received an overall positive rating with four Weak Accept.

The reviewers agree on the effectiveness of the proposed method, the newly contributed dataset, the extensive experimental evaluations, and the writing quality. Clear contributions could be made to the community, as the reviewers appreciated.
There were some concerns regarding the weaknesses of the paper, including concerns around the claims with unclear justifications, method complexity, multi-stage curriculum, significance of the performance, novelty and the suitability of the venue. After the rebuttal and the author-reviewer discussion phase, most concerns were well addressed, as acknowledged by the reviewers.
While the AC agrees that this paper might fit better in a Benchmark/Dataset venue, overall the paper is technically sound, well-written, and would be of interest to at least some fraction of the ICML community.

As a result, the AC is happy to recommend acceptance of this paper, while the authors are asked to incorporate the revisions and the necessary additional justifications in the rebuttal/discussion to their final version.